# Know When to Abstain: Optimal Selective Classification with Likelihood Ratios

**Alvin Heng[1] & Harold Soh[1,2]**
[1]Department of Computer Science, National University of Singapore
[2]Smart Systems Institute, National University of Singapore
`{alvinh, harold}@comp.nus.edu.sg`

## Abstract

Selective classification enhances the reliability of predictive models by allowing them to abstain from making uncertain predictions. In this work, we revisit the design of optimal selection functions through the lens of the Neyman–Pearson lemma, a classical result in statistics that characterizes the optimal rejection rule as a likelihood ratio test. We show that this perspective not only unifies the behavior of several post-hoc selection baselines, but also motivates new approaches to selective classification which we propose here. A central focus of our work is the setting of covariate shift, where the input distribution at test time differs from that at training. This realistic and challenging scenario remains relatively underexplored in the context of selective classification. We evaluate our proposed methods across a range of vision and language tasks, including both supervised learning and vision-language models. Our experiments demonstrate that our Neyman–Pearson-informed methods consistently outperform existing baselines, indicating that likelihood ratio-based selection offers a robust mechanism for improving selective classification under covariate shifts. Our code is publicly available at https://github.com/clear-nus/sc-likelihood-ratios.

## 1 Introduction

Machine learning models are inherently fallible and can make erroneous predictions. Unlike humans, who can abstain from answering when uncertain, e.g., by saying "I don't know", predictive models typically produce a prediction for every input regardless of confidence. Selective classification aims to address this limitation by enabling models to abstain on uncertain inputs, thereby improving overall performance and robustness, for instance by deferring ambiguous cases to human experts.

A wide range of methods have been proposed to determine whether a model should accept or reject an input. Common post-hoc approaches rely on heuristic confidence estimates, such as the maximum softmax probability (Geifman & El-Yaniv, 2017; Hendrycks & Gimpel, 2017), logit margins (Liang et al., 2024), or Monte Carlo dropout (Geifman & El-Yaniv, 2017; Gal & Ghahramani, 2016). Other techniques assess a sample's proximity to the training distribution (Lee et al., 2018; Sun et al., 2022), under the assumption that samples farther from the data manifold are more likely to be misclassified. A separate line of work trains models with explicit abstention mechanisms, such as a rejection logit or head (Geifman & El-Yaniv, 2019; Liu et al., 2019; Huang et al., 2020). In this work we focus on post-hoc methods, which are model-agnostic and do not require specialized training.

Despite the rich literature, two important gaps remain. First, while foundational results (such as Chow (1970); Geifman & El-Yaniv (2017)) provide theoretical underpinnings for selective classification, there is a lack of general, principled guidance for designing effective selector functions in the context of modern deep networks. Second, most evaluations are conducted in the i.i.d. setting, where test data is assumed to follow the training distribution. Few works have begun exploring selective classification under distribution shifts (Xia & Bouganis, 2022; Narasimhan et al., 2024), but focus on the common semantic shifts, neglecting the covariate shift setting that is becoming increasingly relevant.

To address these challenges, we propose a new perspective rooted in the Neyman–Pearson lemma, a classical result from statistics that defines the optimal hypothesis test in terms of a likelihood ratio.

We show that existing selectors can be interpreted as approximations to this test, and we use this insight to derive two new selectors, $\Delta$-MDS and $\Delta$-KNN, as well as a simple linear combination strategy. We evaluate our methods on a comprehensive suite of vision and language benchmarks under covariate shift, where the input distribution changes while the label space remains fixed. We focus on covariate shift for two key reasons: first, it is underexplored relative to semantic shifts (Heng & Soh, 2025) (which is well studied in the context of out-of-distribution detection (Hendrycks & Gimpel, 2017; Ming et al., 2022; Lee et al., 2018; Heng et al., 2024)); second, it is increasingly relevant in modern applications such as vision-language models (VLMs), where the label set is large and variable, rendering most practical shifts in deployment covariate in nature. Our results demonstrate that the proposed selectors outperform existing baselines and provide robust performance across distribution shifts, including on powerful VLMs like CLIP.

In summary, our key contributions are:

1. We introduce for the first time a Neyman–Pearson-based framework for defining optimality in selective classification via likelihood ratio tests.

2. We unify several existing selector methods and propose two new selectors and a linear combination approach under this framework.

3. We conduct a thorough evaluation under distribution shifts, both covariate and semantic, across vision and language tasks and demonstrate superior performance across both VLMs and traditional supervised models.

## 2 BACKGROUND

**Selective Classification.** Consider a standard classification problem with input space $\mathcal{X} \subseteq \mathbb{R}^d$, label space $\mathcal{Y} = \{1, ..., K\}$, and data distribution $\mathcal{D}_{\mathcal{X}, \mathcal{Y}}$ over $\mathcal{X} \times \mathcal{Y}$. A *selective classifier* is a pair $(f, g)$, where $f : \mathcal{X} \to \mathbb{R}^K$ is a base classifier, and $g : \mathcal{X} \to \{0, 1\}$ is a *selector function* that determines whether to make a prediction or abstain. Formally,

$$(f, g)(\boldsymbol{x}) \triangleq \begin{cases} f(\boldsymbol{x}) & \text{if } g(\boldsymbol{x}) = 1, \\ \text{abstain} & \text{if } g(\boldsymbol{x}) = 0. \end{cases} \tag{1}$$

That is, the model abstains on input $\boldsymbol{x}$ when $g(\boldsymbol{x}) = 0$. In practice, $g$ is typically implemented by thresholding a real-valued confidence score:

$$g_{s,\gamma}(\boldsymbol{x}) = \mathbb{1}[s(\boldsymbol{x}) > \gamma], \tag{2}$$

where $s : \mathcal{X} \to \mathbb{R}$ is a confidence scoring function (often adapted from OOD detection methods; see below), and $\gamma$ is a tunable threshold. The performance of a selective classifier is typically evaluated using two metrics:

$$\text{Coverage:} \quad \phi_{s,\gamma} = \mathbb{E}_{\boldsymbol{x} \sim \mathcal{D}_{\mathcal{X}, \mathcal{Y}}}[g_{s,\gamma}(\boldsymbol{x})], \tag{3}$$

$$\text{Selective Risk:} \quad R_{s,\gamma} = \frac{\mathbb{E}_{(\boldsymbol{x}, y) \sim \mathcal{D}_{\mathcal{X}, \mathcal{Y}}}[\ell(f(\boldsymbol{x}), y) \cdot g_{s,\gamma}(\boldsymbol{x})]}{\phi_{s,\gamma}}, \tag{4}$$

where $\ell(f(\boldsymbol{x}), y)$ is the 0/1 loss (Geifman & El-Yaniv, 2017). Selective classification aims to optimize the tradeoff between selective risk and coverage by ideally reducing risk while maintaining high coverage. Improvements can stem from enhancing the base classifier $f$, or from refining the selector $g$ to better identify error-prone inputs. In this work, we fix $f$ to be a strong pretrained model and focus on designing more effective selector functions $g$.

**Covariate Shift.** Covariate shift refers to a scenario where the marginal distribution over inputs, $p(\boldsymbol{x})$, changes between training and testing, while the support of the label distribution $p(y)$ remains unchanged. For example, a model trained on photographs of cats may face a covariate shift when evaluated on paintings of cats, as the input appearance changes but the semantic categories are preserved. This is in contrast to *semantic shift*, where both $p(\boldsymbol{x})$ and $p(y)$ change, typically due to the introduction of unseen classes. In this work, we focus on covariate shifts, which are increasingly relevant in modern applications such as VLMs, where the label set is large and can be adjusted to suit a given task. In such settings, distributional changes primarily manifest as covariate shifts, making them a critical yet underexplored challenge for robust selective classification.

**Out-of-Distribution (OOD) Detection.** OOD detection is closely related to selective classification, as many selector functions $s(\boldsymbol{x})$ are derived from or inspired by OOD scoring methods. Given an in-distribution (ID) data distribution $p_{ID}$, the goal of OOD detection is to construct a scoring function $s : \mathcal{X} \to \mathbb{R}$ such that $s(\boldsymbol{x})$ indicates the likelihood that $\boldsymbol{x}$ originates from $p_{ID}$. A higher $s(\boldsymbol{x})$ corresponds to higher confidence that $\boldsymbol{x}$ is in-distribution. In selective classification, these scores are thresholded to determine whether to accept or abstain on a given input, as formalized in Eq. 2.

## 3 SELECTIVE CLASSIFICATION VIA THE NEYMAN–PEARSON LEMMA

We begin by framing selective classification within the paradigm of hypothesis testing. Let $\mathcal{H}_0 : \mathcal{C}$ denote the hypothesis that the classifier makes a correct prediction, and $\mathcal{H}_1 : \neg\mathcal{C}$ that it makes an incorrect one. Selective classification then reduces to deciding, for each input, whether to accept $\mathcal{H}_0$ or reject in favor of $\mathcal{H}_1$, i.e., a binary decision problem. This perspective is a natural fit for a foundational result from statistics: the *Neyman–Pearson (NP) lemma* (Neyman & Pearson, 1933; Lehmann et al., 1986), which characterizes the optimal decision rule between two competing hypotheses.

**Lemma 1** (Neyman–Pearson (Neyman & Pearson, 1933; Lehmann et al., 1986)). *Let $Z \in \mathbb{R}^d$ be a random variable, and consider the hypotheses::*

$$\mathcal{H}_0 : Z \sim P_0 \quad vs. \quad \mathcal{H}_1 : Z \sim P_1,$$

*where $P_0$ and $P_1$ have densities $p_0$ and $p_1$ that are strictly positive on a shared support $\mathcal{Z} \subset \mathbb{R}^d$. For any measurable acceptance region $A \subset \mathcal{Z}$ under $\mathcal{H}_0$, define the type I error (false rejection) as $\alpha(A) = P_0(Z \notin A)$, and type II error (false acceptance) as $\beta(A) = P_1(Z \in A)$.*

*Fix a type I error tolerance $\alpha_0 \in [0, 1]$. Let $\gamma(\alpha_0)$ be the threshold such that*

$$A^*(\alpha_0) := \left\{ z \in \mathcal{Z} : \frac{p_0(z)}{p_1(z)} \geq \gamma(\alpha_0) \right\}$$

*satisfies $\alpha(A^*) = \alpha_0$. Then $A^*(\alpha_0)$ minimizes the type II error:*

$$\beta(A^*(\alpha_0)) = \min_{A:\alpha(A)=\alpha_0} \beta(A).$$

*In other words, among all decision rules with the same false rejection rate, the likelihood ratio test minimizes the false acceptance rate.*

Applied to selective classification, Lemma 1 suggests the optimal selection score is a *likelihood ratio*:

$$s(\boldsymbol{x}) = \frac{p_c(\boldsymbol{x})}{p_w(\boldsymbol{x})},$$

where $p_c(\boldsymbol{x})$ and $p_w(\boldsymbol{x})$ denote the probability density associated with the classifier making a correct and wrong prediction respectively. Thresholding this score yields the lowest possible selective risk for any given coverage level.

**Corollary 1** (Informal). *Any selector score $s(\boldsymbol{x})$ that is a monotonic transformation of the likelihood ratio $\frac{p_0(\boldsymbol{x})}{p_1(\boldsymbol{x})}$ is also optimal under the Neyman–Pearson criterion.*

Corollary 1 follows directly from the lemma, since monotonic transformations (e.g., logarithmic or affine maps) preserve the ordering of scores and hence do not alter the resulting acceptance region. We define a score function $s(\boldsymbol{x})$ to be *Neyman–Pearson optimal* if it is a monotonic transformation of the likelihood ratio $p_c(\boldsymbol{x})/p_w(\boldsymbol{x})$. In practice, the true likelihood ratio is not accessible, so we approximate it or construct a monotonic proxy that captures the posterior odds of a correct versus incorrect prediction for a given input.

Note that $p_c(\boldsymbol{x})$ and $p_w(\boldsymbol{x})$ are general and naturally accounts for distribution shifts; $p_c(\boldsymbol{x})$ $(p_w(\boldsymbol{x}))$ includes all samples that the classifier classifies correctly (wrongly), *regardless of whether they are ID or a distribution shift*. Therefore, this simplifies our framework compared to prior works that consider ID and OOD distributions separately (Xia & Bouganis, 2022; Narasimhan et al., 2024).

In what follows, we first reinterpret existing selection scores from the literature as implicit approximations to this likelihood ratio and show conditions under which they are NP optimal. We then introduce

two new distance-based selection functions inspired by the NP framework. Finally, we propose a hybrid score that linearly combines multiple selectors which we find performs well in practice. Throughout, the assumptions used in our theoretical results are meant to clarify the connection to NP optimality and the structure of an optimal selector, rather than to prescribe conditions that must hold in practice. Introduction and brief details of the scores discussed in this section are provided in Appendix A.

## 3.1 Logit-Based Scores as Approximations to Likelihood Ratios

We consider two popular confidence scores, Maximum Softmax Probability (MSP) (Hendrycks & Gimpel, 2017) and Raw Logits (RLog) (Liang et al., 2024), and interpret them as approximations to likelihood ratio tests in the NP sense. This provides theoretical justification for their empirical success as selector scores in selective classification.

Let $l^{(1)} \geq l^{(2)} \geq \cdots \geq l^{(K)}$ denote the logits output by a classifier for a sample $\boldsymbol{x}$, sorted in descending order. Define the corresponding softmax probabilities $d^{(i)} = \text{softmax}(l^{(i)})$. The MSP score is given by $s_{\text{MSP}}(\boldsymbol{x}) = d^{(1)}$, while the RLog score is defined as $s_{\text{RLog}}(\boldsymbol{x}) = l^{(1)} - l^{(2)}$. MSP has become a standard baseline for OOD detection and selective classification (Hendrycks & Gimpel, 2017; Geifman & El-Yaniv, 2017), and RLog has been recently proposed as a strong score for selective classification (Liang et al., 2024).

**Theorem 1.** *Let $\hat{y}(x) = \arg\max_{k \in \{1,\ldots,K\}} p_\theta(y = k \mid x)$ be the predicted label and define the event $C := \{\hat{y}(X) = Y\}$ that the classifier is correct. Assume the classifier is calibrated for top-1 correctness, i.e., $P(C \mid X = \boldsymbol{x}) = \max_k p_\theta(y = k \mid \boldsymbol{x}) =: d^{(1)}(\boldsymbol{x})$. Then MSP is Neyman–Pearson optimal for selective classification. Moreover, under the additional assumption that the softmax distribution is concentrated on the top two classes, i.e., $L := \sum_{i \geq 3} d^{(i)} \ll d^{(2)}$, the RLog score is also Neyman–Pearson optimal.*

The proof provided in Appendix B shows that under these assumptions, both MSP and RLog are monotonic transformations of the likelihood ratio $p_c/p_w$, and therefore is NP optimal by Corollary 1. Of course, these assumptions are not always satisfied in practice. Prior work (Guo et al., 2017) has shown that modern neural classifiers tend to be poorly calibrated, and has proposed post-hoc calibration methods such as temperature scaling. Notably, RLog has been shown to be invariant to temperature scaling (Liang et al., 2024), making it robust to miscalibration and a compelling choice in practice. This aligns with our empirical findings in Sec. 5, where RLog generally outperforms MSP (which corresponds to temperature scaling with $T = 1$). While the effect of calibration is an important factor in logit-based methods (Cattelan & Silva, 2023; Fisch et al., 2022), it lies beyond the scope of this work.

## 3.2 Neyman–Pearson Optimal Distance Scores

The logit-based scores discussed in the previous section rely on classifier logit calibration, a condition often violated in practice (Guo et al., 2017). To avoid this dependency, we consider distance-based methods that make alternative assumptions independent of calibration. As we show below, these methods approximate the likelihood ratio $p_c/p_w$ by leveraging spatial relationships in feature space.

Two distance methods widely used in OOD detection are the Mahalanobis distance (MDS) (Lee et al., 2018) and $k$-Nearest Neighbors (KNN) (Sun et al., 2022). Both rely on computing distances between a test sample and training features (see Appendix A for details). Briefly, MDS is defined as $s_{\text{MDS}}(\boldsymbol{x}) = \max_i -(\phi(\boldsymbol{x}) - \mu_i)^\top \Sigma^{-1} (\phi(\boldsymbol{x}) - \mu_i)$, where $\phi(\boldsymbol{x})$ denotes the extracted feature of $\boldsymbol{x}$, typically from the penultimate or final layer of a trained deep network, $\mu_i$ is the empirical mean feature of class $i$, and $\Sigma$ is a shared covariance matrix. In contrast, KNN scores inputs by the negative distance to the $k$-th nearest training feature vector. We introduce $\Delta$-MDS and $\Delta$-KNN, which are modified versions of these scores that explicitly incorporate insights from the NP lemma by estimating separate distributions for correctly and incorrectly classified training samples. Figure 1 gives an overview of our approach, and we provide pseudocode for our proposed methods in Appendix D.

**$\Delta$-MDS.** Instead of estimating a single distribution per class, we maintain two sets of statistics per class: $\{\mu_i^c, \Sigma^c\}_{i=1}^K$ and $\{\mu_i^w, \Sigma^w\}_{i=1}^K$, corresponding to the mean and shared covariance of features for training samples that the classifier predicts correctly and wrongly, respectively. These quantities

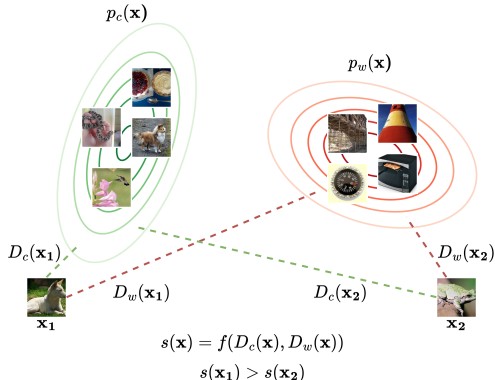

Figure 1: Illustration of our proposed Neyman–Pearson optimal distance-based selective classification methods. We estimate the likelihoods of correct and incorrect predictions ($p_c$ and $p_w$) as a function of distances to training sets consisting of correctly and incorrectly classified samples: $s(\boldsymbol{x}) = f(D_c(\boldsymbol{x}), D_w(\boldsymbol{x}))$, where $f$ here denotes a function. For example, $\boldsymbol{x}_1$ is "closer" to $p_c$ and "farther" from $p_w$ than $\boldsymbol{x}_2$, and should therefore receive a higher score.

are easily estimated as the true labels are known. We then define the $\Delta$-MDS score as the difference in Mahalanobis distances between the two distributions:

$$s_{\Delta\text{-MDS}}(\boldsymbol{x}) = D_{\text{MDS}}(\boldsymbol{x}; \mu_i^c, \Sigma^c) - D_{\text{MDS}}(\boldsymbol{x}; \mu_i^w, \Sigma^w) \tag{5}$$

where $D_{\text{MDS}}(\boldsymbol{x}; \mu_i, \Sigma) = \max_i -(\phi(\boldsymbol{x}) - \mu_i)^\top \Sigma^{-1} (\phi(\boldsymbol{x}) - \mu_i)$ is the standard Mahalanobis score. The score intuitively increases when the input is closer (in Mahalanobis sense) to the "correctly classified" region and farther from the "wrongly classified" region in feature space. We now formalize this intuition:

**Theorem 2.** *Let $Z = \phi(\boldsymbol{x}) \in \mathbb{R}^d$ be the feature representation of input $\boldsymbol{x}$. Let $\mathcal{C}$ be the event the classifier makes a correct prediction and $\neg\mathcal{C}$ its negation. Assume $Z|\mathcal{C} \sim p_c = \mathcal{N}(\mu_i^c, \Sigma^c)$ and $Z|\neg\mathcal{C} \sim p_w = \mathcal{N}(\mu_i^w, \Sigma^w)$. Then the $\Delta$-MDS score $s_{\Delta\text{-MDS}}(\boldsymbol{x})$ is Neyman–Pearson optimal for selective classification.*

The proof is provided in Appendix B, which shows that $s_{\Delta\text{-MDS}}$ is a monotonic transformation of the likelihood ratio $p_c/p_w$, and thus is NP optimal as per Corollary 1. The Gaussian assumption on feature representations is supported both empirically and theoretically via connections between Gaussian Discriminant Analysis and softmax classifiers (Lee et al., 2018), making $\Delta$-MDS well-suited for modern deep classifiers trained on standard supervised learning objectives.

**$\Delta$-KNN.** Next we introduce $\Delta$-KNN, a non-parametric distance-based score inspired by the NP framework. Let $A_c = \{\phi_c(\boldsymbol{x}_1), \ldots, \phi_c(\boldsymbol{x}_{N_c})\}$ and $A_w = \{\phi_w(\boldsymbol{x}_1), \ldots, \phi_w(\boldsymbol{x}_{N_w})\}$ denote the feature representations of training samples that the classifier predicted correctly and wrongly respectively, and $N_c = |A_c|$ and $N_w = |A_w|$. Let $\boldsymbol{z} = \phi(\boldsymbol{x})$ be the feature vector of a test input $\boldsymbol{x}$. Define $u_k(\boldsymbol{z})$ and $v_k(\boldsymbol{z})$ as the Euclidean distances from $\boldsymbol{z}$ to its $k$-th nearest neighbors in $A_c$ and $A_w$. We define the $\Delta$-KNN score as the difference in log-distances:

$$s_{\Delta\text{-KNN}}(\boldsymbol{x}) = D_{\text{KNN}}(\boldsymbol{x}; A_c) - D_{\text{KNN}}(\boldsymbol{x}; A_w) \tag{6}$$

where $D_{\text{KNN}}(\boldsymbol{x}; A_c) = -\log[u_k(\phi(\boldsymbol{x}))]$ and $D_{\text{KNN}}(\boldsymbol{x}; A_w) = -\log[v_k(\phi(\boldsymbol{x}))]$. This score measures how much closer a test point is to the region of correctly classified samples compared to incorrectly classified ones. We now show that $\Delta$-KNN is asymptotically NP optimal:

**Theorem 3.** *Let $Z = \phi(\boldsymbol{x}) \in \mathbb{R}^d$ be the feature representation of input $\boldsymbol{x}$, and let $\mathcal{C}$ denote the event that the classifier makes a correct prediction. Suppose $Z \mid \mathcal{C} \sim p_c$ and $Z \mid \neg\mathcal{C} \sim p_w$ are arbitrary continuous densities bounded away from zero. Let $N_c = |A_c|$ and $N_w = |A_w|$. If $k \to \infty$ while $k/N_c \to 0$ and $k/N_w \to 0$ as $N_c, N_w \to \infty$, then $s_{\Delta\text{-KNN}}(\boldsymbol{x})$ is a Neyman–Pearson optimal selector.*

The proof is provided in Appendix B. As in previous cases, it relies on showing that $s_{\Delta\text{-KNN}}$ is a monotonic transformation of the likelihood ratio $p_c/p_w$. Importantly, this result does not require parametric assumptions on the form of $p_c$ or $p_w$, unlike $\Delta$-MDS. However, it does depend on

asymptotic properties of the $k$-nearest neighbor density estimator, and the required conditions on $k$, $N_c$, and $N_w$ may be difficult to satisfy in finite-sample settings. As such, both methods have their tradeoffs in terms of modeling assumptions.

In practice, we replace the single $k$-th neighbor distance with the average log-distance to the top $k$ neighbors. Specifically, we use: $D_{\text{KNN}}(\boldsymbol{x}; A_c) = -\frac{1}{k} \sum_{i=1}^{k} \log[u_i(\phi(\boldsymbol{x}))]$ and $D_{\text{KNN}}(\boldsymbol{x}; A_w) = -\frac{1}{k} \sum_{i=1}^{k} \log[v_i(\phi(\boldsymbol{x}))]$. We find that this smoother version improves empirical performance, as shown in our ablation studies in Sec. 5.3. While this modification deviates from the form in Theorem 3, we include a discussion in Appendix C suggesting NP optimality holds for the averaged log-distance formulation under standard assumptions.

### 3.3 Linear Combinations Of Distance and Logit-based Scores

The selector scores we discussed rely on different modeling assumptions and exhibit complementary strengths, as discussed in Wang et al. (2022). Logit-based scores utilize the classifier's learned boundaries, while distance-based methods depend on geometric structures in feature space defined by training samples. We are thus motivated to leverage their respective advantages by proposing a simple yet effective solution: linearly combining selector scores. Intuitively, this allows each score to compensate for the limitations of the other. The following lemma formalizes the NP optimality of such a linear combination:

**Lemma 2.** *Let $s_1(\boldsymbol{x}) \in \mathbb{R}$ and $s_2(\boldsymbol{x}) \in \mathbb{R}$ be two selector scores. Assume both are Neyman–Pearson optimal; that is, $s_1(\boldsymbol{x})$ is a monotone transform of $p_c^{(1)}/p_w^{(1)}$ and $s_2(\boldsymbol{x})$ is a monotone transform of $p_c^{(2)}/p_w^{(2)}$. Then for any scalar $\lambda \in \mathbb{R}$, $t(\boldsymbol{x}) = s_1(\boldsymbol{x}) + \lambda s_2(\boldsymbol{x})$ is a monotonic transformation of $p_c^{(1)}(p_c^{(2)})^\lambda/p_w^{(1)}(p_w^{(2)})^\lambda$.*

The proof provided in Appendix B follows by expressing $t(\boldsymbol{x})$ as a log-product of likelihood ratios. Thus, $t(\boldsymbol{x})$ remains NP optimal under the assumption that the density for each hypothesis takes the form of a multiplicative (or "tilted") product: $p_c^{(1)}(p_c^{(2)})^\lambda/Z_c$ and similarly for $p_w$, where $Z_c$ is a normalization constant. In practice, we find that combining a distance-based score (e.g., $\Delta$-MDS) with a logit-based score (e.g., RLog) leads to the best performance. We refer to such combinations by concatenating their names, e.g., $\Delta$-MDS-RLog. We discuss fitting parameters like $\lambda$ in Sec. 5.

## 4 Related Works

The study of classification with a reject option has a long history, beginning with cost-based formulations (Chow, 1970) and extensions to classical models like SVMs (Fumera & Roli, 2002; Bartlett & Wegkamp, 2008) and nearest neighbors (Hellman, 1970). In deep learning, LeCun et al. (1989) explored rejection via top logit activations. Later, the risk–coverage and classifier–selector frameworks were formalized (El-Yaniv et al., 2010; Geifman & El-Yaniv, 2017), with methods like MSP and Monte Carlo dropout proposed to provide confidence-based selection (Gal & Ghahramani, 2016).

Subsequent works have extended this direction by studying popular logit and distance-based scores, many originally developed for OOD detection. Examples include MSP (Hendrycks & Gimpel, 2017), MaxLogit (Hendrycks et al., 2019), Energy (Liu et al., 2020), MDS (Lee et al., 2018), and KNN (Sun et al., 2022). A limitation of logit-based methods is their reliance on classifier calibration. Although calibration is not the focus of our work, several studies have examined its impact on selective classification performance (Cattelan & Silva, 2023; Galil et al., 2023). An alternative line of research uses conformal prediction to construct calibrated prediction sets with formal guarantees (Vovk et al., 2005; Angelopoulos & Bates, 2021; Bates et al., 2021; Angelopoulos et al., 2024). While such methods could be adapted for selective classification, they differ fundamentally from our goal of designing scoring functions optimized for selective risk. Jiang et al. (2018) proposes a trust score by comparing the model's prediction with class-conditional KNN distances to estimate if a sample will be correctly classified, but it was shown to be ineffective on high-dimensional images.

Some methods incorporate rejection directly into training. For instance, SelectiveNet (Geifman & El-Yaniv, 2019) adds a dedicated rejection head, while Deep Gamblers (Liu et al., 2019) and Self-Adaptive Training (Huang et al., 2020) introduce a reject class and train the model to abstain. These methods require architectural modifications and joint training. In contrast, our work focuses on post-hoc methods that can be applied to pretrained classifiers without retraining.

Table 1: DFN CLIP AURC (A) and NAURC (N) results on ImageNet and its covariate shifted variants at full coverage. Lower is better. AURC results are on the $10^{-2}$ scale. **Bold** and underline denotes the best and second best result respectively. "Avg (1K)" denotes average results over datasets with full 1K-class coverage, while "Avg (all)" is the average result over all datasets.

| Method | Im-1K A | Im-1K N | Im-R A | Im-R N | Im-A A | Im-A N | ON A | ON N | Im-V2 A | Im-V2 N | Im-S A | Im-S N | Im-C A | Im-C N | Avg (1K) A | Avg (1K) N | Avg (all) A | Avg (all) N |
|---|---|---|---|---|---|---|---|---|---|---|---|---|---|---|---|---|---|---|
| MSP | 9.08 | 0.542 | 2.00 | 0.344 | 2.29 | 0.179 | 8.87 | 0.268 | 9.39 | 0.521 | 12.3 | 0.524 | 15.1 | 0.328 | 11.5 | 0.479 | 8.43 | 0.387 |
| MaxLogit | 9.08 | 0.542 | 2.21 | 0.385 | 2.96 | 0.249 | 12.4 | 0.437 | 9.35 | 0.518 | 12.3 | 0.525 | 17.7 | 0.423 | 13.9 | 0.502 | 11.3 | 0.440 |
| Energy | 14.2 | 0.901 | 5.81 | 1.06 | 12.2 | 1.2 | 33.3 | 1.43 | 14.9 | 0.882 | 20.8 | 0.975 | 49.2 | 1.59 | 24.8 | 1.09 | 21.5 | 1.15 |
| MDS | 11.3 | 0.699 | 3.41 | 0.608 | 3.21 | 0.274 | 16.4 | 0.624 | 11.7 | 0.672 | 15.2 | 0.68 | 17.6 | 0.426 | 13.9 | 0.619 | 11.3 | 0.569 |
| KNN | 10.5 | 0.643 | 2.51 | 0.439 | 2.67 | 0.219 | 11.4 | 0.390 | 10.9 | 0.618 | 14.1 | 0.619 | 16.7 | 0.389 | 13.1 | 0.567 | 9.83 | 0.474 |
| RLog | 4.83 | 0.246 | 0.808 | 0.122 | 1.59 | 0.108 | 7.73 | 0.214 | 5.27 | 0.250 | 6.84 | 0.234 | 12.6 | 0.226 | 7.39 | 0.239 | 5.67 | 0.200 |
| SIRC | 16.3 | 1.04 | 3.90 | 0.701 | 8.48 | 0.817 | 15.1 | 0.564 | 17.1 | 1.03 | 20.7 | 0.968 | 22.1 | 0.612 | 19.05 | 0.913 | 14.8 | 0.819 |
| Δ-MDS | 5.00 | 0.257 | 2.19 | 0.380 | 2.43 | 0.194 | 9.63 | 0.304 | 5.43 | 0.260 | 8.28 | 0.311 | 12.5 | 0.224 | 7.81 | 0.263 | 6.50 | 0.276 |
| Δ-KNN | 4.60 | 0.230 | 1.42 | 0.237 | 1.99 | 0.149 | 8.52 | 0.252 | 4.99 | 0.231 | 7.55 | 0.272 | 12.1 | 0.207 | 7.32 | 0.235 | 5.89 | 0.225 |
| Δ-MDS-RLog | 4.13 | 0.197 | 1.09 | 0.175 | 1.60 | 0.109 | **7.13** | **0.185** | 4.52 | 0.200 | 6.27 | 0.204 | **11.1** | **0.170** | 6.51 | 0.193 | 5.12 | 0.177 |
| Δ-KNN-RLog | **3.98** | **0.187** | **0.770** | **0.115** | **1.45** | **0.093** | 7.14 | 0.186 | **4.36** | **0.190** | **6.13** | **0.196** | 11.3 | 0.175 | **6.43** | **0.187** | **5.01** | **0.163** |

Related to our work are approaches that combine selective classification and OOD detection, termed SCOD (Xia & Bouganis, 2022; Narasimhan et al., 2024). Such methods consider the ID classification and OOD distributions separately and seek to combine them into a single score function. These approaches are typically designed for semantic shifts and need adaptation for covariate shift. Our formulation avoids this by representing all distribution shifts through the general pair $(p_c, p_w)$, which does not require distinguishing between shift types.

Finally, most closely related to our work is Liang et al. (2024), who study selective classification under both semantic and covariate shifts and introduce the Raw Logit (RLog) score. Our work differs in several ways: 1) we focus on covariate shifts, which we argue is more relevant in modern settings where large and variable label sets (e.g., from vision-language models) mitigate label drift; 2) we introduce a unified theoretical framework grounded in the Neyman–Pearson lemma, from which we derive new selector scores with formal optimality guarantees; and 3) we evaluate our methods on a broader class of models, including VLMs, whereas Liang et al. (2024) focus exclusively on standard supervised learning paradigms.

## 5 EXPERIMENTS

**Datasets.** We evaluate our methods across vision and language domains, with a primary focus on the former. For vision tasks, we use ImageNet-1K (Im-1K) and a suite of covariate-shifted variants: 1) ImageNet-Rendition (Im-R) (Hendrycks et al., 2020), 2) ImageNet-A (Im-A) (Hendrycks et al., 2021), 3) ObjectNet (ON) (Barbu et al., 2019), 4) ImageNetV2 (Im-V2) (Recht et al., 2019), 5) ImageNet-Sketch (Im-S) (Wang et al., 2019), and 6) ImageNet-C (Im-C) (Hendrycks & Dietterich, 2019). We group these datasets based on label coverage: full 1000-class coverage (Im-1K, Im-V2, Im-S, Im-C) and subsets of classes (Im-R, Im-A, ON). For language tasks, we evaluate on the Amazon Reviews dataset (Ni et al., 2019; Koh et al., 2021). To simulate realistic deployment scenarios involving distribution shift, following Liang et al. (2024) we evaluate on mixed test sets that combine in-distribution and covariate-shifted samples. For example, results reported on Im-C are computed on a combined test set of Im-1K and Im-C.

**Classifiers and Baseline Selector Scores.** We consider two families of classifiers for vision experiments, namely CLIP zero-shot VLMs (Radford et al., 2021) and supervised classifiers. Specifically, we use the CLIP model from Data Filtering Networks (DFN) (Fang et al., 2024) and EVA (Fang et al., 2023) for supervised learning, chosen for their state-of-the-art accuracy on ImageNet. Our focus is not on model training but on evaluating selector scores applied post-hoc. Note that for EVA, we restrict evaluation to datasets with full 1K class coverage as the model is trained on the complete ImageNet label set only. In contrast, CLIP can be adapted at inference time to arbitrary label subsets, so we evaluate it across all datasets. For language tasks, we fine-tune a DistilBERT (Sanh et al., 2019) model using LISA (Yao et al., 2022) on the Amazon Reviews training set and evaluate selective classification performance on the full test set.

Table 3: Supervised learning AURC (A) and NAURC (N) results with the EVA model at full coverage. Lower is better. AURC results are on the $10^{-2}$ scale. **Bold** and underline denotes the best and second best result respectively.

| Method | Im-1K | | Im-V2 | | Im-S | | Im-C | | Avg (1K) | |
|---|---|---|---|---|---|---|---|---|---|---|
| | A | N | A | N | A | N | A | N | A | N |
| MSP | 3.32 | 0.256 | 3.85 | 0.266 | 8.15 | 0.319 | 6.41 | 0.215 | 5.43 | 0.264 |
| MaxLogit | 4.53 | 0.371 | 5.16 | 0.379 | 10.3 | 0.437 | 7.98 | 0.301 | 6.99 | 0.372 |
| Energy | 6.82 | 0.590 | 7.58 | 0.589 | 14.0 | 0.641 | 11.1 | 0.474 | 9.89 | 0.573 |
| MDS | 4.01 | 0.322 | 4.32 | 0.307 | 7.26 | 0.271 | 6.80 | 0.236 | 5.60 | 0.284 |
| KNN | 4.00 | 0.321 | 4.31 | 0.306 | 7.15 | 0.265 | 6.77 | 0.234 | 5.56 | 0.282 |
| RLog | 2.33 | 0.161 | 2.72 | 0.168 | 5.90 | 0.197 | 5.50 | 0.163 | 4.11 | 0.172 |
| SIRC | 3.68 | 0.290 | 4.23 | 0.299 | 8.71 | 0.350 | 6.84 | 0.240 | 5.87 | 0.295 |
| $\Delta$-MDS | 2.56 | 0.183 | 2.90 | 0.183 | 5.76 | 0.189 | 5.50 | 0.164 | 4.18 | 0.180 |
| $\Delta$-KNN | 2.60 | 0.187 | 2.99 | 0.191 | 5.91 | 0.197 | 5.74 | 0.177 | 4.31 | 0.188 |
| $\Delta$-MDS-RLog | **2.26** | **0.155** | **2.61** | **0.158** | **5.45** | **0.172** | **5.12** | **0.143** | **3.86** | **0.157** |
| $\Delta$-KNN-RLog | 2.31 | 0.159 | 2.69 | 0.165 | 5.63 | 0.182 | 5.38 | 0.157 | 4.00 | 0.166 |

For baseline scores, we compare our proposed $\Delta$-MDS, $\Delta$-KNN, and their linear combinations with common OOD detection and uncertainty-based scores: MSP (Hendrycks & Gimpel, 2017), MCM (for CLIP), MaxLogit (Hendrycks et al., 2019), Energy (Liu et al., 2020), MDS (Lee et al., 2018), KNN (Sun et al., 2022), and RLog (Liang et al., 2024) as well as SIRC (Xia & Bouganis, 2022). As they are functionally similar, we abbreviate MCM as MSP when presenting CLIP results. Details of the baseline are provided in Appendix A.

**Evaluation Metrics.** We evaluate performance using two metrics: the Area Under the Risk-Coverage Curve (AURC) and the Normalized AURC (NAURC) (Cattelan & Silva, 2023). The AURC captures the joint performance of the classifier and selector across coverage levels. NAURC normalizes AURC to account for the classifier's base error rate, providing a fairer comparison across models with different accuracies. Formally, NAURC is defined as:

$$\text{NAURC}(f, g) = \frac{\text{AURC}(f, g) - \text{AURC}(f, g^*)}{R(f) - \text{AURC}(f, g^*)}, \tag{7}$$

where $g^*$ denotes an oracle confidence function achieving optimal AURC, and $R(f)$ is the risk of $f$. The oracle can be computed in practice using the ground-truth labels of the evaluation set. Intuitively, NAURC measures how close the selector $g$ gets to the optimal, normalized by the classifier's total error. Thus, while AURC is useful for understanding overall performance in the context of a specific model, NAURC enables fair selector comparisons across models by factoring out baseline classifier accuracy.

**Selecting $\lambda$ and $k$.** Both $\lambda$ and $k$ can be selected on a validation set. We found that the simplest recipe to fitting $\lambda$ is to balance the magnitudes of $s_1(\boldsymbol{x})$ and $s_2(\boldsymbol{x})$, so that neither overpowers the other. For $k$ in KNN-based scores, we find that $k \in [25, 50]$ is a sweet spot. Full experimental settings are provided in Appendix E, and hyperparameter sensitivity analysis for $\lambda$ and $k$ is presented in appendix Fig. 2.

Table 2: Results on Amazon Reviews and its covariate shifted test set at full coverage using Distil-BERT trained with LISA.

| Method | In-D | | Cov Shift | |
|---|---|---|---|---|
| | A | N | A | N |
| MSP | 12.2 | 0.368 | 13.9 | 0.401 |
| MaxLogit | 12.6 | 0.384 | 14.3 | 0.416 |
| Energy | 12.89 | 0.397 | 14.6 | 0.428 |
| MDS | 20.6 | 0.739 | 22.2 | 0.750 |
| KNN | 19.4 | 0.686 | 21.3 | 0.711 |
| RLog | 12.4 | 0.376 | 14.1 | 0.410 |
| SIRC | 12.3 | 0.370 | 14.0 | 0.403 |
| $\Delta$-MDS | 12.7 | 0.389 | 14.4 | 0.422 |
| $\Delta$-KNN | 12.4 | 0.374 | 14.2 | 0.412 |
| $\Delta$-MDS-RLog | 12.2 | 0.368 | 13.9 | 0.401 |
| $\Delta$-KNN-RLog | 12.0 | 0.358 | 13.8 | 0.394 |
| $\Delta$-MDS-MSP | **11.9** | **0.354** | **13.6** | **0.387** |
| $\Delta$-KNN-MSP | 12.0 | 0.359 | 13.8 | 0.396 |

## 5.1 IMAGE EXPERIMENTS

We report full selective classification results for CLIP and EVA models in Table 1 and Table 3, respectively. First, let us consider CLIP results. We see that going from MDS and KNN to their NP-informed variants, $\Delta$-MDS and $\Delta$-KNN, leads to roughly 50% reduction in average AURC and NAURC, showing that the assumptions made in the NP-optimality theory hold well in practice. The best average performance is achieved by the linear combinations $\Delta$-KNN-RLog and $\Delta$-MDS-RLog, with $\Delta$-KNN-RLog leading overall in both AURC and NAURC. RLog score ranks third on average, highlighting its strength as a standalone

Table 4: Ablation experiments on DFN CLIP.

| Method | Im-1K | | Im-R | | Im-A | | ON | | Im-V2 | | Im-S | | Im-C | | Avg (1K) | | Avg (all) | |
|---|---|---|---|---|---|---|---|---|---|---|---|---|---|---|---|---|---|---|
| | A | N | A | N | A | N | A | N | A | N | A | N | A | N | A | N | A | N |
| Ablations on $\Delta$-KNN | | | | | | | | | | | | | | | | | | |
| $\Delta$-KNN no avg | 4.66 | 0.234 | **1.40** | **0.234** | 2.16 | 0.167 | 8.87 | 0.268 | 5.11 | 0.239 | 7.63 | 0.276 | 12.4 | 0.216 | 7.45 | 0.241 | 6.03 | 0.233 |
| $\Delta$-KNN w/ avg | **4.60** | **0.230** | 1.42 | 0.237 | **1.99** | **0.149** | **8.52** | **0.252** | 4.99 | 0.231 | 7.55 | 0.272 | 12.1 | 0.207 | 7.32 | 0.235 | 5.89 | 0.225 |
| Ablations on linear combinations | | | | | | | | | | | | | | | | | | |
| $\Delta$-MDS-$\Delta$-KNN | 4.68 | 0.235 | 1.98 | 0.237 | 2.26 | 0.149 | 9.06 | 0.252 | 5.11 | 0.231 | 7.90 | 0.272 | 12.2 | 0.207 | 7.46 | 0.235 | 6.16 | 0.225 |
| MSP-RLog | 4.82 | 0.245 | 0.800 | 0.120 | 1.56 | 0.105 | 7.51 | 0.204 | 5.26 | 0.249 | 6.81 | 0.232 | 12.5 | 0.222 | 7.35 | 0.237 | 5.61 | 0.197 |
| $\Delta$-KNN-MSP | 4.57 | 0.228 | 1.22 | 0.199 | 1.82 | 0.131 | 7.58 | 0.207 | 4.94 | 0.228 | 7.40 | 0.264 | 11.8 | 0.210 | 7.18 | 0.244 | 5.62 | 0.253 |
| $\Delta$-KNN-RLog | **3.98** | **0.187** | **0.770** | **0.115** | **1.49** | **0.093** | **7.14** | **0.186** | **4.36** | **0.190** | **6.13** | **0.196** | **11.3** | **0.175** | **6.43** | **0.187** | **5.01** | **0.163** |

logit-based selector. Motivated by this strong performance, we use RLog in combination with our distance-based scores. We plot the risk-coverage curves for selected datasets in Fig. 3 of the appendix, showing our methods consistently demonstrate the most favorable trade-off across all coverage levels, remaining stable even at low coverage.

For practitioners aiming to identify the best overall selective classification setup that considers both the base classifier and the selector, one approach is to compare performance using the AURC metric. On the 1K-class datasets, EVA paired with $\Delta$-MDS-RLog achieves an AURC of 3.86, outperforming the DFN CLIP model with $\Delta$-KNN-RLog at 5.01. Despite similar NAURC values, EVA's higher Im-1K base accuracy (84.33% vs. DFN CLIP's 80.39%) makes it the preferred choice when considering both components. Intuitively, the optimal setup involves pairing the best selector (here, $\Delta$-MDS-RLog) with the most accurate base classifier.

For EVA, the ranking is reversed: $\Delta$-MDS-RLog achieves the best overall performance, followed by $\Delta$-KNN-RLog. This supports our hypothesis that MDS-based methods are particularly effective for supervised models due to the close connection between softmax classifiers and Gaussian Discriminant Analysis (Lee et al., 2018), which justifies the Gaussian assumptions used in MDS. In contrast, CLIP models trained with contrastive learning (Radford et al., 2021) do not satisfy these assumptions, making the nonparametric $\Delta$-KNN combination more suitable. The bottom row of Fig. 3 of the appendix confirms that $\Delta$-MDS-RLog yields the best risk-coverage behavior for supervised learning across all coverage levels. Comparing average NAURC on the full 1K-class datasets, $\Delta$-MDS-RLog with EVA is the top performing selector score, with a slightly better score (0.157) than the best performer on CLIP (0.163).

To verify that the performance gains of our methods stem from actual algorithmic improvements, rather than large-scale pretraining of CLIP or EVA potentially mitigating distribution shifts, we also evaluate on ResNet50 trained solely on ImageNet-1K. The results in appendix Table 7 show that our methods perform the best, consistent with earlier findings.

**Semantic Shift Experiments.** For completeness, we also report results for experiments on datasets that are semantic shifts to ImageNet-1K in Appendix Table 8. In agreement with the covariate shift experiments, our proposed methods achieve the best performance on this benchmark.

## 5.2 LANGUAGE EXPERIMENTS

Table 2 presents results on the Amazon Reviews dataset. Unlike the vision tasks, the best-performing method is $\Delta$-MDS-MSP, followed closely by $\Delta$-MDS-RLog and $\Delta$-KNN-MSP. Since LISA (Yao et al., 2022) uses a softmax classification objective, the superiority of MDS-based selectors supports our hypothesis about their suitability for supervised models. Interestingly, MSP outperforms RLog in this domain, resulting in better performance when combined with $\Delta$-MDS. This highlights another important practical insight that the best linear combination often involves pairing the top-performing standalone distance-based and logit-based score.

Table 5: Ablation results using DFN-CLIP on ImageNet-1K where the fraction of labeled samples used in feature computation for our proposed methods are varied.

| | 0.1% | | 1% | | 10% | | 50% | | 100% | |
|---|---|---|---|---|---|---|---|---|---|---|
| Method | A | N | A | N | A | N | A | N | A | N |
| $\Delta$-MDS-RLog | - | - | 10.5 | 0.638 | 4.19 | 0.202 | 4.14 | 0.198 | 4.13 | 0.197 |
| $\Delta$-KNN-RLog | 4.81 | 0.245 | 4.58 | 0.229 | 4.17 | 0.200 | 3.98 | 0.188 | 3.98 | 0.187 |

## 5.3 ABLATIONS

**Design Choices.** Table 4 summarizes several ablation experiments on the design choices of our proposed methods. First, we justify averaging the top-$k$ nearest neighbor distances in $\Delta$-KNN rather than using the $k$-th distance alone. This modification yields measurable gains, where average AURC improves from 6.03 to 5.89 and NAURC improves from 0.233 to 0.225. We also investigate various combinations of selector scores. For CLIP, $\Delta$-KNN-RLog remains the best across all configurations, outperforming both double-distance combinations (e.g., $\Delta$-MDS-$\Delta$-KNN) and double-logit combinations (e.g., MSP-RLog). Notably, pairing $\Delta$-KNN with RLog significantly outperforms pairing it with MSP, further validating RLog's role as a strong logit-based complement.

**Sample Efficiency.** Although our methods require a one-time feature computation step, this cost is amortized over all future inference runs as the resulting features can be cached. Nevertheless, to evaluate performance in low-data or low-computation resource regimes, we conducted ablations limiting the amount of labeled samples used. The results in Table 5 show that both methods are surprisingly stable. $\Delta$-KNN is especially robust, maintaining strong performance with as little as 0.1% of labeled data. As expected, $\Delta$-MDS degrades at the 1% level due to the difficulty of estimating per-class statistics with so few samples, and is not applicable at 0.1% (roughly 1 image per class). Importantly, $\Delta$-KNN-RLog continues to outperform RLog at 1% and matches it at 0.1% (see Table 1), indicating that our method is still preferable whenever even a small amount of labeled data is accessible.

## 6 CONCLUSION

We presented a framework for designing selector functions for selective classification, grounded in the Neyman–Pearson lemma. This reveals that the optimal selection score is a monotonic transformation of a likelihood ratio, which unifies several existing methods. We proposed two novel distance-based scores and their linear combinations with logit-based baselines. Experiments across vision and language demonstrate that our methods achieve state-of-the-art performance across diverse settings.

**Limitations and Future Work.** While our focus has been on classification, the Neyman–Pearson framework is general and broadly applicable to other predictive tasks. Exploring selective prediction in settings where uncertainty plays a critical role, such as semantic segmentation and time series forecasting, presents promising future directions. Additionally, extending these ideas to generative models such as LLMs is another exciting avenue for future work.

ACKNOWLEDGEMENTS

This research / project is supported by A*STAR under its National Robotics Programme (NRP) (Award M23NBK0053). The authors also acknowledge support from Google (Google South Asia and South-East Asia Award).

ETHICS STATEMENT

This work does not involve human subjects, and it relies solely on publicly available models and datasets with all attributions provided. Based on the scope of our methods and results, we do not identify ethical issues that require special attention, and we are not aware of immediate harmful applications arising from this research.

REPRODUCIBILITY STATEMENT

We include the theoretical and experimental details necessary to reproduce our findings. Proofs and supporting discussions for all theoretical claims appear in appendix Sec. B and Sec. C. For implementation clarity, we provide pseudocode for our methods in appendix Sec. D and report experimental setup and hyperparameters in appendix Sec. E.

LLM USAGE

We used large language models to assist in formulating and checking mathematical proofs and to improve grammar and writing clarity. All outputs from the LLMs were reviewed by the authors before inclusion, and the authors take full responsibility for the paper's content and the accuracy of the presented results.

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

# Appendix for "Know When to Abstain: Optimal Selective Classification with Likelihood Ratios"

## A  DESCRIPTION OF BASELINES

In this section we provide a brief description of each baseline considered in this work.

**Maximum Softmax Probability (MSP) (Hendrycks & Gimpel, 2017).**  Given an input $\boldsymbol{x}$, let the classifier output logits be $\{l^{(k)}\}_{k=1}^{K}$ and corresponding softmax probabilities $p_\theta(y = k|\boldsymbol{x}) = \text{softmax}(l^{(k)})$. The MSP score is defined as

$$s_{\text{MSP}}(\boldsymbol{x}) = \max_{k \in \{1,\ldots,K\}} p_\theta(y = k|\boldsymbol{x}).$$

MSP is commonly used as an OOD/confidence score: larger values indicate the model places high probability mass on a single class, and thus the input is treated as more "in-distribution" or "confident". For selective classification, we threshold this score (and subsequent scores) to form the selector $g_{s,\gamma}(\boldsymbol{x}) = \mathbb{1}[s_{\text{MSP}}(\boldsymbol{x}) > \gamma]$.

**Maximum Logit (MaxLogit) (Hendrycks et al., 2019).**  The MaxLogit score is defined as

$$s_{\text{MaxLogit}}(\boldsymbol{x}) = \max_{k \in \{1,\ldots,K\}} l^{(k)}$$

Intuitively, larger values indicate that at least one class is assigned a large unnormalized score, and thus the model is more confident in its prediction.

**Energy (Liu et al., 2020).**  The energy score is defined from logits as the Helmholtz free energy

$$E(\boldsymbol{x}) = -T \log \sum_{k=1}^{K} \exp\left(l^{(k)}/T\right),$$

where $T > 0$ is a temperature parameter (we set $T = 1$). For selective classification we use the negative energy (so that larger values indicate more in-distribution-like inputs):

$$s_{\text{Energy}}(\boldsymbol{x}) = \log \sum_{k=1}^{K} \exp\left(l^{(k)}\right).$$

**Mahalanobis Distance (MDS) (Lee et al., 2018).**  Let $\boldsymbol{z} = \phi(\boldsymbol{x})$ denote the feature representation of $\boldsymbol{x}$ (e.g., penultimate features of a deep network). Using training data, we estimate the empirical mean feature $\mu_i$ for each class $i \in \{1,\ldots,K\}$ and a shared (tied) covariance matrix $\Sigma$ across classes. The MDS score is then

$$s_{\text{MDS}}(\boldsymbol{x}) = \max_{i \in \{1,\ldots,K\}} -(\boldsymbol{z} - \mu_i)^\top \Sigma^{-1}(\boldsymbol{z} - \mu_i),$$

i.e., the negative squared Mahalanobis distance to the closest class centroid. Intuitively, $s_{\text{MDS}}(\boldsymbol{x})$ is large when $\boldsymbol{x}$ lies in a high-density region of some class under the fitted Gaussian discriminant model, and small when $\boldsymbol{x}$ is far from all class clusters.

**$k$-Nearest Neighbors (KNN) (Sun et al., 2022).**  Let $\{\boldsymbol{z}_j\}_{j=1}^{n}$ denote the set of training features (we normalize the features). Define $r_k(\boldsymbol{z})$ as the Euclidean distance from $\boldsymbol{z}$ to its $k$-th nearest neighbor among $\{\boldsymbol{z}_j\}$. The KNN score is

$$s_{\text{KNN}}(\boldsymbol{x}) = -r_k(\boldsymbol{z}),$$

so that inputs lying in denser regions of the training feature manifold (smaller neighbor distances) receive higher scores.

**Raw Logits (RLog) (Liang et al., 2024).** RLog uses the confidence margin between the top two logits as a scale-robust confidence score. Let $l^{(1)} \geq l^{(2)} \geq \cdots \geq l^{(k)}$ be the logits sorted in descending order for a given input $\boldsymbol{x}$. We define

$$s_{\mathrm{RLog}}(\boldsymbol{x}) \;=\; l^{(1)} - l^{(2)}$$

Intuitively, $s_{\mathrm{RLog}}(\boldsymbol{x})$ is large when there is a clear "winner" class, and small when the classifier is uncertain between competing labels. Liang et al. (2024) argue that using a logit-space margin yields a more robust score under classifier miscalibration and post-hoc logit transformations (e.g., temperature scaling), since it depends only on the relative separation between the top classes.

**Softmax Information Retaining Combination (SIRC) (Xia & Bouganis, 2022).** Let $S_1(\boldsymbol{x})$ be a primary softmax-derived confidence score, and let $S_2(\boldsymbol{x})$ be an auxiliary feature-based score. We choose $S_1$ to be MSP and $S_2$ to be the $L_1$ norm, in line with Xia & Bouganis (2022). SIRC combines these via

$$s_{\mathrm{SIRC}}(\boldsymbol{x}) \;=\; -\big(S_1^{\mathrm{max}} - S_1(\boldsymbol{x})\big)\Big(1 + \exp\big(-b\,[S_2(\boldsymbol{x}) - a]\big)\Big),$$

where $S_1^{\mathrm{max}}$ is the maximum attainable value of $S_1$ (e.g., $S_1^{\mathrm{max}} = 1$ for MSP), and $a, b$ control how strongly $S_2$ influences the score. We follow Xia & Bouganis (2022) and set $a$ and $b$ using in-distribution statistics of $S_2$ ($a = \mu_{S_2} - 3\sigma_{S_2}$ and $b = 1/\sigma_{S_2}$).

## B  PROOFS

**Theorem 1.** *Let $\hat{y}(x) = \arg\max_{k \in \{1,\dots,K\}} p_\theta(y = k \mid x)$ be the predicted label and define the event $C := \{\hat{y}(X) = Y\}$ that the classifier is correct. Assume the classifier is calibrated for top-1 correctness, i.e., $P(C \mid X = \boldsymbol{x}) = \max_k p_\theta(y = k \mid \boldsymbol{x}) =: d^{(1)}(\boldsymbol{x})$. Then MSP is Neyman–Pearson optimal for selective classification. Moreover, under the additional assumption that the softmax distribution is concentrated on the top two classes, i.e., $L := \sum_{i \geq 3} d^{(i)} \ll d^{(2)}$, the RLog score is also Neyman–Pearson optimal.*

*Proof.* Recall that we denote $l^{(1)} \geq \cdots \geq l^{(K)}$ as the logits predicted by the classifier for a given input $\boldsymbol{x}$ (sorted in descending order) and $d^{(i)} = \mathrm{softmax}(l^{(i)})$ the corresponding softmax probabilities. Then $s_{\mathrm{MSP}}(\boldsymbol{x}) = d^{(1)}(\boldsymbol{x})$ and $s_{\mathrm{RLog}}(\boldsymbol{x}) = l^{(1)}(\boldsymbol{x}) - l^{(2)}(\boldsymbol{x})$.

Define

$$p_c(\boldsymbol{x}) := p(\boldsymbol{x} \mid C) \quad \text{and} \quad p_w(\boldsymbol{x}) := p(\boldsymbol{x} \mid \neg C),$$

i.e., the (test-time) input densities conditioned on the classifier being correct or wrong, respectively. Let $\pi := P(C)$ and define the posterior correctness probability $q(\boldsymbol{x}) := P(C \mid X = \boldsymbol{x})$.

**MSP Optimality.** By Bayes' rule,

$$q(\boldsymbol{x}) = \frac{p(\boldsymbol{x} \mid C)\, P(C)}{p(\boldsymbol{x} \mid C)\, P(C) + p(\boldsymbol{x} \mid \neg C)\, P(\neg C)}$$

$$= \frac{p_c(\boldsymbol{x})\, \pi}{p_c(\boldsymbol{x})\, \pi + p_w(\boldsymbol{x})\, (1 - \pi)}.$$

Equivalently, the posterior odds satisfy

$$\frac{q(\boldsymbol{x})}{1 - q(\boldsymbol{x})} = \frac{p_c(\boldsymbol{x})}{p_w(\boldsymbol{x})} \cdot \frac{\pi}{1 - \pi}.$$

Since the classifier is (perfectly) calibrated for top-1 correctness such that $d^{(1)}(\boldsymbol{x}) = P(C \mid X = \boldsymbol{x}) = q(\boldsymbol{x})$,

$$s_{\mathrm{MSP}}(\boldsymbol{x}) = d^{(1)}(\boldsymbol{x}) = q(\boldsymbol{x}) = \frac{\frac{\pi}{1-\pi} \frac{p_c(\boldsymbol{x})}{p_w(\boldsymbol{x})}}{1 + \frac{\pi}{1-\pi} \frac{p_c(\boldsymbol{x})}{p_w(\boldsymbol{x})}}. \tag{8}$$

The mapping $h(z) = \frac{z}{1+z}$ is strictly increasing for $z \geq 0$ since $h'(z) = \frac{1}{(1+z)^2} > 0$, and scaling by the positive constant $\frac{\pi}{1-\pi}$ preserves monotonicity. Hence $s_{\mathrm{MSP}}(\boldsymbol{x})$ is a monotone transformation of the likelihood ratio $\frac{p_c(\boldsymbol{x})}{p_w(\boldsymbol{x})}$, and is Neyman–Pearson optimal by Corollary 1.

**RLog Optimality.** We can express RLog as the logarithm of the ratio of the top two softmax values:

$$\frac{d^{(1)}(x)}{d^{(2)}(\boldsymbol{x})} = \frac{e^{l^{(1)}(\boldsymbol{x})}}{e^{l^{(2)}(\boldsymbol{x})}} = e^{l^{(1)}(\boldsymbol{x}) - l^{(2)}(\boldsymbol{x})} = e^{s_{\mathrm{RLog}}(\boldsymbol{x})},$$

thus $s_{\mathrm{RLog}}(\boldsymbol{x}) = \log \frac{d^{(1)}(\boldsymbol{x})}{d^{(2)}(\boldsymbol{x})}$. Observe that

$$\frac{d^{(1)}(\boldsymbol{x})}{d^{(2)}(\boldsymbol{x})} = \frac{d^{(1)}(\boldsymbol{x})}{1 - d^{(1)}(\boldsymbol{x})} \left(1 + \frac{L(\boldsymbol{x})}{d^{(2)}(\boldsymbol{x})}\right), \tag{9}$$

where $L(x) = 1 - d^{(1)}(\boldsymbol{x}) - d^{(2)}(\boldsymbol{x}) = \sum_{i \geq 3} d^{(i)}(\boldsymbol{x}) \geq 0$. In the binary classification case, $L(\boldsymbol{x}) = 0$ and hence

$$s_{\mathrm{RLog}}(\boldsymbol{x}) = \log \frac{d^{(1)}(\boldsymbol{x})}{1 - d^{(1)}(\boldsymbol{x})} = \log \frac{q(\boldsymbol{x})}{1 - q(\boldsymbol{x})} = \log \frac{p_c(\boldsymbol{x})}{p_w(\boldsymbol{x})} + \log \frac{\pi}{1 - \pi},$$

which differs from $\log \frac{p_c(\boldsymbol{x})}{p_w(\boldsymbol{x})}$ only by an additive constant. Since $\log(\cdot)$ is strictly increasing and additive constants do not change ordering, $s_{\mathrm{RLog}}$ is Neyman–Pearson optimal by Corollary 1.

In the multiclass case, the extra term $\log\left(1 + \frac{L(\boldsymbol{x})}{d^{(2)}(\boldsymbol{x})}\right)$ may vary across samples and can in principle affect ordering. Under the stated assumption $L(x) \ll d^{(2)}(\boldsymbol{x})$, which is empirically supported by high top-5 classification accuracies in prior works (Liu et al., 2021; Brock et al., 2021), this term is small and varies little, so $s_{\mathrm{RLog}}(\boldsymbol{x}) \approx \log \frac{q(\boldsymbol{x})}{1 - q(\boldsymbol{x})}$ and remains an approximately monotone proxy for $\log \frac{p_c(\boldsymbol{x})}{p_w(\boldsymbol{x})}$, yielding (approximate) Neyman–Pearson optimality. □

**Theorem 2.** *Let $Z = \phi(\boldsymbol{x}) \in \mathbb{R}^d$ be the feature representation of input $\boldsymbol{x}$. Let $\mathcal{C}$ be the event the classifier makes a correct prediction and $\neg\mathcal{C}$ its negation. Assume $Z|\mathcal{C} \sim p_c = \mathcal{N}(\mu_i^c, \Sigma^c)$ and $Z|\neg\mathcal{C} \sim p_w = \mathcal{N}(\mu_i^w, \Sigma^w)$. Then the $\Delta$-MDS score $s_{\Delta\text{-MDS}}(\boldsymbol{x})$ is Neyman–Pearson optimal for selective classification.*

*Proof.* The likelihood of a multivariate Gaussian in $\mathbb{R}^d$ is $p(z; \mu, \Sigma) = (2\pi)^{-d/2} \det(\Sigma)^{-1/2} \exp\left(-\frac{1}{2}(x - \mu)^\top \Sigma^{-1}(x - \mu)\right)$.

We see that the Mahalanobis distance $D(z; \mu, \Sigma)$ is proportional to the log-likelihood of the multivariate Gaussian. As such, assuming that the underlying $p_c$ and $p_w$ follow multivariate Gaussians of the form $\mathcal{N}(\mu_i^c, \Sigma^c)$ and $\mathcal{N}(\mu_i^w, \Sigma^w)$ respectively,

$$s_{\Delta\text{-MDS}}(\boldsymbol{x}) = D_{\mathrm{MDS}}(\boldsymbol{x}; \mu_i^c, \Sigma^c) - D_{\mathrm{MDS}}(\boldsymbol{x}; \mu_i^w, \Sigma^w)$$
$$= 2\log \frac{p_c(\boldsymbol{z}; \mu_i^c, \Sigma^c)}{p_w(\boldsymbol{z}; \mu_i^w, \Sigma^w)} + \log \frac{\det \Sigma^c}{\det \Sigma^w}.$$

Therefore, $s_{\Delta\text{-MDS}}(\boldsymbol{x})$ is a monotone transform of $p_c/p_w$ and is Neyman–Pearson optimal by Corollary 1.

□

**Theorem 3.** *Let $Z = \phi(\boldsymbol{x}) \in \mathbb{R}^d$ be the feature representation of input $\boldsymbol{x}$, and let $\mathcal{C}$ denote the event that the classifier makes a correct prediction. Suppose $Z \mid \mathcal{C} \sim p_c$ and $Z \mid \neg\mathcal{C} \sim p_w$ are arbitrary continuous densities bounded away from zero. Let $N_c = |A_c|$ and $N_w = |A_w|$. If $k \to \infty$ while $k/N_c \to 0$ and $k/N_w \to 0$ as $N_c, N_w \to \infty$, then $s_{\Delta\text{-KNN}}(\boldsymbol{x})$ is a Neyman–Pearson optimal selector.*

*Proof.* The empirical likelihood of the KNN density estimator (Silverman, 2018; Zhao & Lai, 2022) is given by

$$\hat{p}_c(\boldsymbol{z}) = \frac{k}{N_c V_d(u_k(\boldsymbol{z}))^d}, \quad \hat{p}_w(\boldsymbol{z}) = \frac{k}{N_w V_d(v_k(\boldsymbol{z}))^d}, \tag{10}$$

where $k \geq 2$, $u_k(\boldsymbol{z})$ and $v_k(\boldsymbol{z})$ are the Euclidean distances from $\boldsymbol{z}$ to its $k$-th nearest neighbor from $A_c$ and $A_w$ and $V_d$ is the unit-ball volume in $\mathbb{R}^d$. A classic result of non-parametric nearest neighbor density estimation (Loftsgaarden & Quesenberry, 1965) states that as $k \to \infty$ but $k/N_c \to 0$,

$k/N_w \to 0$, then $\hat{p}_c(\boldsymbol{z}) \to p_c(\boldsymbol{z})$ and $\hat{p}_w(\boldsymbol{z}) \to p_w(\boldsymbol{z})$ for every $\boldsymbol{z}$. In other words, the empirical KNN density estimator converges to the true density under the stated asymptotic conditions.

One can see that the difference in log-likelihoods is

$$
\begin{aligned}
&\log \hat{p}_c(\boldsymbol{z}) - \log \hat{p}_w(\boldsymbol{z}) \\
&\qquad = -d \log u_k(\boldsymbol{z}) + d \log v_k(\boldsymbol{z}) + \log \frac{N_w}{N_c}.
\end{aligned} \tag{11}
$$

Therefore

$$
\begin{aligned}
s_{\Delta\text{-KNN}}(\boldsymbol{z}) &\triangleq -\log u_k(\boldsymbol{z}) + \log v_k(\boldsymbol{z}) \\
&= \frac{1}{d} \log \frac{\hat{p}_c(\boldsymbol{z})}{\hat{p}_w(\boldsymbol{z})} - \frac{1}{d} \log \frac{N_w}{N_c}.
\end{aligned} \tag{12}
$$

Since the last term is constant, $s_{\Delta\text{-KNN}}(\boldsymbol{z})$ is a monotone transform of $\frac{\hat{p}_c(\boldsymbol{z})}{\hat{p}_w(\boldsymbol{z})}$. Under the stated conditions on $k$, $N_c$ and $N_w$, the empirical likelihoods converge to the true likelihoods $p_c$ and $p_w$, thus $s_{\Delta\text{-KNN}}(\boldsymbol{z})$ is also Neyman–Pearson optimal. $\qquad\square$

**Lemma 2.** *Let $s_1(\boldsymbol{x}) \in \mathbb{R}$ and $s_2(\boldsymbol{x}) \in \mathbb{R}$ be two selector scores. Assume both are Neyman–Pearson optimal; that is, $s_1(\boldsymbol{x})$ is a monotone transform of $p_c^{(1)}/p_w^{(1)}$ and $s_2(\boldsymbol{x})$ is a monotone transform of $p_c^{(2)}/p_w^{(2)}$. Then for any scalar $\lambda \in \mathbb{R}$, $t(\boldsymbol{x}) = s_1(\boldsymbol{x}) + \lambda s_2(\boldsymbol{x})$ is a monotonic transformation of $p_c^{(1)}(p_c^{(2)})^\lambda / p_w^{(1)}(p_w^{(2)})^\lambda$.*

*Proof.* Let

$$
L_1(\boldsymbol{x}) = \frac{p_c^{(1)}(\boldsymbol{x})}{p_w^{(1)}(\boldsymbol{x})}, \quad L_2(\boldsymbol{x}) = \frac{p_c^{(2)}(\boldsymbol{x})}{p_w^{(2)}(\boldsymbol{x})}. \tag{13}
$$

Since each score is already a strictly monotone transform of $L(\boldsymbol{x})$, we are free to re-express the scores in any other convenient monotone scale without affecting relative ordering and thus Neyman–Pearson optimality. Without loss of generality, we will let $s_i(\boldsymbol{x}) = \log \frac{p_c^{(i)}(\boldsymbol{x})}{p_w^{(i)}(\boldsymbol{x})}, i = 1, 2$, which are identical to the original scores in terms of sample acceptance and rejection patterns. Given $\lambda \in \mathbb{R}$,

$$
\begin{aligned}
t(\boldsymbol{x}) &= s_1(\boldsymbol{x}) + \lambda s_2(\boldsymbol{x}) \tag{14} \\
&= \log L_1(\boldsymbol{x}) + \lambda \log L_2(\boldsymbol{x}) \tag{15} \\
&= \log(L_1(\boldsymbol{x}) L_2(\boldsymbol{x})^\lambda) \tag{16} \\
&= \log \frac{p_c^{(1)}(p_c^{(2)})^\lambda}{p_w^{(1)}(p_w^{(2)})^\lambda} \tag{17}
\end{aligned}
$$

In other words, $t(\boldsymbol{x})$ is a monotone transform of the tilted likelihood ratio $p_c^{(1)}(p_c^{(2)})^\lambda / p_w^{(1)}(p_w^{(2)})^\lambda$. $\qquad\square$

Recall $\mathcal{H}_0$ and $\mathcal{H}_1$ represent the hypotheses that the classifier will make a correct and wrong prediction respectively. Assuming that the density of $\mathcal{H}_0$ takes the form of a tilted likelihood $p_c^{(1)}(p_c^{(2)})^\lambda / Z_c$, where $Z_c$ is a normalizing constant, and vice-versa for $\mathcal{H}_1$, then $t(\boldsymbol{x})$ is Neyman–Pearson optimal by Corollary 1.

## C  AVERAGE TOP $k$ $\Delta$-KNN MODIFICATION

Here we discuss how Neyman–Pearson optimality of the average log-distance formulation of $\Delta$-KNN can hold as described in the main text. Recall that we let $D_{\text{KNN}}(\boldsymbol{z}; A_c) = -\frac{1}{k} \sum_{i=1}^{k} \log(u_i(\boldsymbol{z}))$ and vice-versa for $A_w$.

For concreteness, let us consider distances to the correct set; the derivation is identical for the wrong set. In the asymptotic limit where $N_c$ is large and $k \ll N_c$, the ball centered at $\boldsymbol{z}$ that just encloses its

$k$th nearest neighbor (i.e., with volume $V_d(u_k(\boldsymbol{z}))^d$) is so small that the true density is essentially constant over it, so the radii of the first $k$ neighbors are *conditionally i.i.d. uniform* in the ball.

As such, let us define the normalized variable

$$U_i = \left(\frac{u_i(\boldsymbol{z})}{u_k(\boldsymbol{z})}\right)^d, \quad i = 1, ..., k. \tag{18}$$

Note that $U_i \in [0, 1]$ for all $i$. Since the joint distribution of $U_i$ depends only on $k$, we know from the $i$-th order statistics of $k$ i.i.d. $\mathrm{Uniform}(0, 1)$ variables that each $U_i$ is Beta-distributed, $U_i \sim \mathrm{Beta}(i, k - i + 1)$, $0 \leq U_1 \leq \cdots \leq U_k = 1$. With some algebra, $\log u_i(\boldsymbol{z}) = \frac{1}{d} \log U_i + \log u_k(\boldsymbol{z})$. Then,

$$\frac{1}{k} \sum_{i=1}^{k} \log(u_i(\boldsymbol{z})) = \log u_k(\boldsymbol{z}) + \frac{1}{kd} \sum_{i=1}^{k} \log U_i \tag{19}$$

The second term converges almost surely to

$$\frac{1}{kd} \sum_{i=1}^{k} \log U_i \to \frac{1}{d} \int_0^1 \log x \, \mathrm{d}x = -\frac{1}{d} \tag{20}$$

as $k \to \infty$ as it is a sum of $k$ i.i.d. $\mathrm{Uniform}(0, 1)$ random variables. In other words, in the asymptotic limit the average log-distance is a monotone transform of the log-distance itself. By substituting Eq. 19 back into Eq. 11 for distances to both correct and wrong sets, the modified $\Delta$-KNN formulation remains a monotone transform o $p_c/p_w$, thus suggesting Neyman-Pearson optimality under Corollary 1.

## D    ALGORITHM PSEUDOCODE FOR PROPOSED SCORES

Pseudocode for Algorithms 1 ($\Delta$-MDS and its linear combination) and 2 (Scoring with $\Delta$-KNN and its linear combination) are shown on the next page.

---

**Algorithm 1** Scoring with $\Delta$-MDS and its linear combination

---

**Input:** Trained classifier $f$, feature extractor $\phi$ (typically penultimate or final layer of $f$), training set $\mathcal{D}_{\text{train}} = \{(\boldsymbol{x}_i, y_i)\}$, test set $\mathcal{D}_{\text{test}} = \{\boldsymbol{x}_j\}$, optional logit-based score function $s_{\text{logit}}(\boldsymbol{x})$, combination weight $\lambda$

**Output:** Selector scores $s(\boldsymbol{x})$ for each $\boldsymbol{x} \in \mathcal{D}_{\text{test}}$

1: Initialize $\mathcal{A}_c^{(i)} \leftarrow \emptyset$ and $\mathcal{A}_w^{(i)} \leftarrow \emptyset$ for $i = 1$ to $K$ $\triangleright$ Class-wise correct and incorrect feature sets
2: **for** each $(\boldsymbol{x}, y)$ in $\mathcal{D}_{\text{train}}$ **do**
3:     $\hat{y} \leftarrow f(\boldsymbol{x})$
4:     $\boldsymbol{z} \leftarrow \phi(\boldsymbol{x})$
5:     **if** $\hat{y} = y$ **then**
6:         Add $\boldsymbol{z}$ to $\mathcal{A}_c^{(y)}$
7:     **else**
8:         Add $\boldsymbol{z}$ to $\mathcal{A}_w^{(y)}$
9:     **end if**
10: **end for**
11: Compute $\{\mu_i^c, \Sigma^c\}_{i=1}^K$ and $\{\mu_i^w, \Sigma^w\}_{i=1}^K$ from $\mathcal{A}_c$ and $\mathcal{A}_w$
12: **for** each $\boldsymbol{x}$ in $\mathcal{D}_{\text{test}}$ **do**
13:     $\boldsymbol{z} \leftarrow \phi(\boldsymbol{x})$
14:     $d_c \leftarrow \max_i -(\boldsymbol{z} - \mu_i^c)^\top (\Sigma^c)^{-1}(\boldsymbol{z} - \mu_i^c)$
15:     $d_w \leftarrow \max_i -(\boldsymbol{z} - \mu_i^w)^\top (\Sigma^w)^{-1}(\boldsymbol{z} - \mu_i^w)$
16:     $s_{\Delta\text{-MDS}}(\boldsymbol{x}) \leftarrow d_c - d_w$
17:     **if** using linear combination **then**
18:         $s(\boldsymbol{x}) \leftarrow s_{\Delta\text{-MDS}}(\boldsymbol{x}) + \lambda \cdot s_{\text{logit}}(\boldsymbol{x})$
19:     **else**
20:         $s(\boldsymbol{x}) \leftarrow s_{\Delta\text{-MDS}}(\boldsymbol{x})$
21:     **end if**
22: **end for**
23: **return** $\{s(\boldsymbol{x})\}$ for each $\boldsymbol{x} \in \mathcal{D}_{\text{test}}$

---

---

**Algorithm 2** Scoring with $\Delta$-KNN and its linear combination

---

**Input:** Trained classifier $f$, feature extractor $\phi$ (typically penultimate or final layer of $f$), training set $\mathcal{D}_{\text{train}} = \{(\boldsymbol{x}_i, y_i)\}$, test set $\mathcal{D}_{\text{test}} = \{\boldsymbol{x}_j\}$, number of neighbors $k$, optional logit-based score function $s_{\text{logit}}(\boldsymbol{x})$, combination weight $\lambda$

**Output:** Selector scores $s(\boldsymbol{x})$ for each $\boldsymbol{x} \in \mathcal{D}_{\text{test}}$

1: Initialize $\mathcal{A}_c \leftarrow \emptyset$, $\mathcal{A}_w \leftarrow \emptyset$            $\triangleright$ Global sets of correct and incorrect features
2: **for** each $(\boldsymbol{x}, y)$ in $\mathcal{D}_{\text{train}}$ **do**
3:      $\hat{y} \leftarrow f(\boldsymbol{x})$
4:      $\boldsymbol{z} \leftarrow \phi(\boldsymbol{x})$
5:      **if** $\hat{y} = y$ **then**
6:          Add $\boldsymbol{z}$ to $\mathcal{A}_c$
7:      **else**
8:          Add $\boldsymbol{z}$ to $\mathcal{A}_w$
9:      **end if**
10: **end for**
11: **for** each $\boldsymbol{x}$ in $\mathcal{D}_{\text{test}}$ **do**
12:      $\boldsymbol{z} \leftarrow \phi(\boldsymbol{x})$
13:      Compute $\{u_i\}_{i=1}^k \leftarrow$ distances from $\boldsymbol{z}$ to $k$ nearest neighbors in $\mathcal{A}_c$
14:      Compute $\{v_i\}_{i=1}^k \leftarrow$ distances from $\boldsymbol{z}$ to $k$ nearest neighbors in $\mathcal{A}_w$
15:      $d_c \leftarrow -\frac{1}{k} \sum_{i=1}^k \log(u_i)$
16:      $d_w \leftarrow -\frac{1}{k} \sum_{i=1}^k \log(v_i)$
17:      $s_{\Delta\text{-KNN}}(\boldsymbol{x}) \leftarrow d_c - d_w$
18:      **if** using linear combination **then**
19:          $s(\boldsymbol{x}) \leftarrow s_{\Delta\text{-KNN}}(\boldsymbol{x}) + \lambda \cdot s_{\text{logit}}(\boldsymbol{x})$
20:      **else**
21:          $s(\boldsymbol{x}) \leftarrow s_{\Delta\text{-KNN}}(\boldsymbol{x})$
22:      **end if**
23: **end for**
24: **return** $\{s(\boldsymbol{x})\}$ for each $\boldsymbol{x} \in \mathcal{D}_{\text{test}}$

---

# E    EXPERIMENTAL DETAILS

**Image Experiments.**    This section outlines the models and datasets used in the vision experiments in Section 5. For classifiers, we use the ViT-H/14 variant for DFN CLIP and the "Giant" variant of EVA, both with patch size 14. Pretrained weights are obtained from the `OpenCLIP`[1] (Cherti et al., 2023) and `timm`[2] libraries, respectively.

ImageNet and its covariate-shifted variants are downloaded from their respective open-source repositories[3][4][5][6][7][8][9]. For ImageNetV2, we use the MatchedFrequency test set to match the frequency distribution of the original ImageNet. For ImageNet-C, we evaluate using corruption level 5 to simulate the most challenging conditions. The dataset includes four corruption categories: 1) blur, 2) digital, 3) noise, and 4) weather. Each category contains multiple corruption types (e.g., Gaussian, impulse, and shot noise under the noise category). To ensure balanced evaluation, we first average results within each category, then average across the four categories to give equal weight to each corruption type.

For logit-based scores with CLIP, we are required to construct logits over class concepts by taking the dot product between the text embedding $T_\theta(c)$ of class concepts $c$ and the image embedding, $\phi(x)$, where $T_\theta$ is the text encoder of CLIP. Given a class label $y$, we construct the class concept with the template "a real, high-quality, clear and clean photo of a $\{y\}$" when computing the confidence scores for logit-based selectors. We found this to improve scores slightly as opposed to using the default template in the original work (Radford et al., 2021). We attribute this to the hypothesis that CLIP should produce lower confidence scores when faced with covariate-shifted inputs, such as sketches or corrupted images, as the error rate on covariate shifts are much higher (Heng & Soh, 2025) than on Im-1K, which are generally clear and well-lit photographs.

Distance-based methods such as MDS, KNN, and our proposed variants compute distances in the feature space. For CLIP, we use the output of the final layer of the vision encoder; for EVA, we use the penultimate layer output.

The hyperparameters $\lambda$ (for linear combinations) and $k$ (for KNN-based scores) used in the experiments reported in Table 1 and Table 3 are summarized in Table 6. All experiments are conducted on a single NVIDIA A6000 GPU with 48GB of memory.

**Language Experiments.**    For language experiments, we fine-tune a DistilBERT model on the Amazon Reviews dataset using the training pipeline provided in the official LISA repository[10], with default hyperparameters. For distance-based selectors, we extract features from the penultimate layer of DistilBERT, consistent with the EVA setting. The values of $\lambda$ and $k$ used in Table 2 are also listed in Table 6. All language experiments are run on a single NVIDIA A6000 48GB GPU.

# F    ADDITIONAL RESULTS

The following provides additional experimental results showing risk-coverage trade-offs, hyperparameter sensitivity, and performance results on both semantic and covariate shift.

---

[1] https://github.com/mlfoundations/open_clip
[2] https://github.com/huggingface/pytorch-image-models
[3] https://www.image-net.org/
[4] https://github.com/hendrycks/imagenet-r
[5] https://github.com/hendrycks/natural-adv-examples
[6] https://objectnet.dev/
[7] https://imagenetv2.org/
[8] https://github.com/HaohanWang/ImageNet-Sketch
[9] https://github.com/hendrycks/robustness
[10] https://github.com/huaxiuyao/LISA

Table 6: Values of $\lambda$ and $k$ for results reported in Sec. 5.

| Method | $\lambda$ | $k$ |
|---|---|---|
| DFN CLIP | | |
| KNN | - | 50 |
| $\Delta$-KNN | - | 25 |
| $\Delta$-MDS-RLog | 10000 | - |
| $\Delta$-KNN-RLog | 10 | 25 |
| Eva | | |
| KNN | - | 50 |
| $\Delta$-KNN | - | 25 |
| $\Delta$-MDS-RLog | 1000 | - |
| $\Delta$-KNN-RLog | 0.5 | 25 |
| DistilBERT | | |
| KNN | - | 50 |
| $\Delta$-KNN | - | 25 |
| $\Delta$-MDS-RLog | 1000 | - |
| $\Delta$-KNN-RLog | 0.05 | 25 |
| $\Delta$-MDS-MSP | 1000 | - |
| $\Delta$-KNN-MSP | 0.5 | 25 |

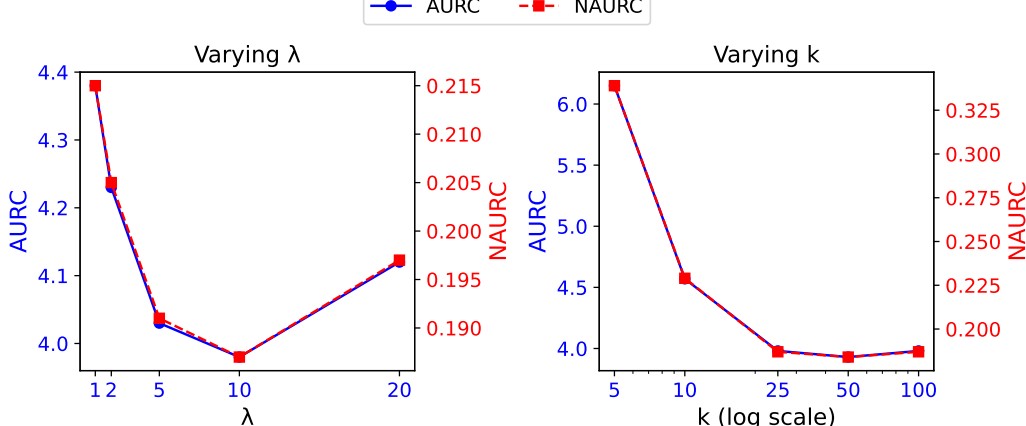

Figure 2: Hyperparameter sensitivity plots for $\lambda$ and $k$ for $\Delta$-KNN-RLog with DFN on ImageNet1K. The results for this combination are not highly sensitive to $\lambda$, while for $k$ results plateau at around $k = 25$.

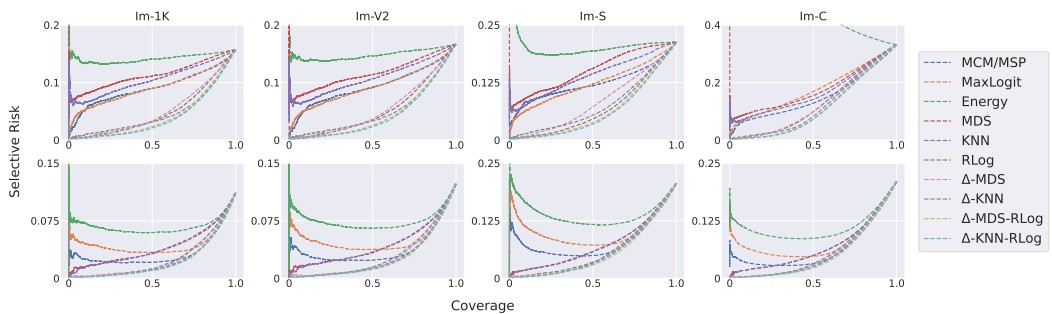

Figure 3: Risk-coverage curves of various selector methods for CLIP (top row) and EVA (bottom row). Our proposed methods consistently achieve the best risk-coverage tradeoff and remain stable at low coverage levels.

Table 7: Additional results under covariate shift using ResNet50 trained on ImageNet1K from the official PyTorch repository. Consistent with the findings on larger models in the main text, our proposed methods outperform all baselines. These results affirm that the gains stem from algorithmic improvements in selective classification, rather than from large-scale pretraining potentially mitigating distribution shifts.

| | Im-1K | | Im-V2 | | Im-S | | Im-C | | Avg (1K) | |
|---|---|---|---|---|---|---|---|---|---|---|
| Method | A | N | A | N | A | N | A | N | A | N |
| MSP | 9.67 | 0.248 | 10.32 | 0.257 | 23.4 | 0.196 | 22.6 | 0.203 | 16.5 | 0.226 |
| MaxLogit | 12.7 | 0.379 | 13.5 | 0.394 | 25.3 | 0.248 | 24.5 | 0.260 | 19.0 | 0.320 |
| Energy | 16.3 | 0.539 | 17.6 | 0.567 | 28.1 | 0.328 | 27.8 | 0.358 | 22.5 | 0.448 |
| MDS | 21.2 | 0.752 | 21.6 | 0.736 | 58.4 | 1.19 | 58.3 | 1.23 | 39.9 | 0.977 |
| KNN | 28.8 | 1.08 | 30.4 | 1.11 | 40.2 | 0.673 | 46.2 | 0.900 | 36.4 | 0.940 |
| RLog | 8.61 | 0.201 | 9.14 | 0.207 | 24.5 | 0.225 | 22.4 | 0.193 | 16.2 | 0.207 |
| SIRC | 18.9 | 0.648 | 19.6 | 0.652 | 46.2 | 0.844 | 40.5 | 0.729 | 31.3 | 0.718 |
| Δ-MDS | 13.0 | 0.392 | 13.7 | 0.399 | 30.3 | 0.390 | 27.3 | 0.338 | 21.1 | 0.380 |
| Δ-KNN | 15.3 | 0.493 | 16.2 | 0.509 | 36.0 | 0.552 | 33.9 | 0.529 | 25.3 | 0.521 |
| Δ-MDS-RLog | **8.33** | **0.189** | **8.86** | **0.195** | **24.1** | 0.215 | **21.9** | **0.180** | **15.8** | **0.195** |
| Δ-KNN-RLog | 8.48 | 0.195 | 9.02 | 0.202 | 24.3 | 0.221 | 22.2 | 0.189 | 16.0 | 0.202 |

Table 8: Additional results on semantic shift datasets, namely ImageNet-O, iNaturalist, SUN and Places. The ID distribution is ImageNet-1K.

| | ImageNet-O | | iNaturalist | | SUN | | Places | | Avg | |
|---|---|---|---|---|---|---|---|---|---|---|
| | A | N | A | N | A | N | A | N | A | N |
| DFN CLIP | | | | | | | | | | |
| MSP | 9.70 | 0.459 | 11.7 | 0.273 | 12.2 | 0.293 | 12.8 | 0.319 | 11.6 | 0.336 |
| MDS | 11.8 | 0.584 | 14.1 | 0.370 | 14.8 | 0.398 | 15.2 | 0.414 | 14.0 | 0.442 |
| MaxLogit | 9.75 | 0.462 | 11.9 | 0.282 | 12.6 | 0.311 | 13.4 | 0.343 | 11.9 | 0.350 |
| Energy | 17.7 | 0.930 | 28.1 | 0.935 | 33.2 | 1.14 | 32.9 | 1.13 | 28.0 | 1.03 |
| KNN | 11.3 | 0.552 | 13.2 | 0.334 | 13.9 | 0.360 | 13.9 | 0.362 | 13.1 | 0.402 |
| RLog | 6.21 | 0.253 | 10.0 | 0.205 | 11.2 | 0.254 | 11.5 | 0.267 | 9.73 | 0.245 |
| SIRC | 18.2 | 0.958 | 32.7 | 1.12 | 30.2 | 1.02 | 32.3 | 1.11 | 28.4 | 1.05 |
| Δ-KNN | 5.93 | 0.237 | 9.42 | 0.181 | 10.6 | 0.228 | 10.4 | 0.220 | 9.09 | 0.217 |
| Δ-KNN-RLog | **5.20** | **0.194** | **8.60** | **0.148** | **9.69** | **0.192** | **9.76** | **0.195** | **8.31** | **0.182** |
| EVA | | | | | | | | | | |
| MSP | 4.94 | 0.285 | 8.46 | 0.214 | 9.29 | 0.251 | 10.1 | 0.289 | 8.20 | 0.260 |
| MDS | 4.58 | 0.258 | 6.64 | 0.133 | 7.41 | 0.167 | 7.78 | 0.184 | 6.60 | 0.186 |
| MaxLogit | 6.37 | 0.392 | 10.8 | 0.319 | 11.5 | 0.349 | 12.6 | 0.399 | 10.3 | 0.365 |
| Energy | 8.80 | 0.572 | 14.5 | 0.487 | 14.5 | 0.484 | 15.8 | 0.545 | 13.4 | 0.522 |
| KNN | 4.63 | 0.262 | 6.70 | 0.135 | 7.43 | 0.168 | 7.70 | 0.180 | 6.62 | 0.186 |
| RLog | 3.82 | 0.201 | 6.92 | 0.145 | 8.25 | 0.205 | 8.21 | 0.203 | 6.80 | 0.189 |
| SIRC | 5.41 | 0.320 | 9.14 | 0.245 | 10.2 | 0.293 | 11.1 | 0.331 | 8.96 | 0.297 |
| Δ-MDS | 3.49 | **0.177** | **5.63** | **0.087** | **6.72** | **0.136** | **6.95** | **0.146** | **5.70** | **0.137** |
| Δ-MDS-RLog | **3.37** | 0.202 | 5.64 | 0.130 | 6.90 | 0.187 | 7.08 | 0.186 | 5.75 | 0.176 |

