# OpenReview forum: "Know When to Abstain: Optimal Selective Classification with Likelihood Ratios"
_ICLR.cc/2026/Conference — ICLR 2026 Poster_

### Official Review · Reviewer_55hp · 2025-10-27

**Soundness:** 2
**Presentation:** 3
**Contribution:** 3
**Rating:** 6
**Confidence:** 3

**Summary:**

The paper considers the problem of abstaining from making a classification decision. Abstaining can be useful when the classifier is likely to make a wrong prediction. The authors consider a formalism where abstaining decision is made by thresholding a confidence scoring function. Next, the authors review the Neyman–Pearson (NP) lemma on the optimal decision, and observe that some of the popular scoring functions can be viewed as approximations of the optimal rule. The authors then introduce two new scoring methods and describe conditions under which the methods are NP optimal. Effectiveness of the proposed scoring functions is supported by experiments on vision and language data.

**Strengths:**

1. The problem of abstaining rather than making incorrect predictions is an important practical problem
2. The authors offer a framework to unify previous and newly proposed confidence scoring functions. Relevance to the NP lemma is an insightful observation
3. The paper provides formal arguments (i.e., proofs) on optimality of different scores
4. Evaluation on different datasets shows usefulness of the proposed scores
5. The paper is clearly presented. There are minor issues, but overall the paper is easy to follow

**Weaknesses:**

1. On several occasions, justification of assumptions and theoretical constructs is not clear. First, it is not clear why p(y) should remain unchanged. It changes if relative frequencies of classes change. Also, it is not clear why exactly this assumption is required. Second, the practical implications of Lemma 2 are not clear. Third, Theorem 1 uses symbol "<<", which informally means "much smaller", but does not have any formal meaning
2. The newly introduced scores are not fundamentally new, since MDS and KNN scores have already been considered
3. On the practical side, it is not clear what amount of labelled data (e.g., relative to the amount of training data) is needed for the method to work reliably. This can be an issue, because modern classifiers can be constructed from pre-trained models with a minimum amount of training data (e.g., using few shot learning).
4. In terms of presentation, it would be useful to define AURC within the paper. Also, I'm not sure how NP Lemma implies that "thresholding this score yields the lowest possible selective risk for any given coverage level"
5. The experiments do not provide confidence intervals or p-values

**Questions:**

1. What amount of labelled data (e.g., relative to the amount of training data) is needed for the method to work reliably?
2. What do assumptions of Theorem 2 mean in practice?
3. What are the practical implications of Lemma 2?

---

> ### Author Response · Authors · 2025-11-19
> **Author response part 1**
>
> We thank the reviewer for acknowledging the strengths of our work and for their constructive feedback. Below we provide our responses to the concerns and questions raised by the reviewer.
>
> > *On several occasions, justification of assumptions and theoretical constructs is not clear. First, it is not clear why p(y) should remain unchanged. It changes if relative frequencies of classes change.*
>
> **Response**: We thank the reviewer for raising this point. The reviewer is correct that under covariate shift, **it is the *support* of $p(y)$ that remains unchanged** (no new labels introduced), not necessarily the distribution $p(y)$ itself. Equivalently, in measure-theoretic terms, the label space $\mathcal{Y}$ is fixed (subsets of $\mathcal{Y}$ are allowed) while the marginal distribution $p(x)$ over $\mathcal{X}$ changes. We will amend this in the revised manuscript. **Importantly, this clarification does not affect our conclusion**: the covariate shift datasets we evaluate on already differ from ImageNet in both label frequencies and label coverage, and our methods operate well under these conditions.
>
> > *Also, it is not clear why exactly this assumption is required.*
>
> **Response**: The reviewer is correct that **our method is general and agnostic to distribution shifts**, which we mentioned in Line 315: *“Our formulation … represents all distribution shifts through the general pair $(p_c, p_w)$, which does not require distinguishing between shift types”.*
>
> As such, the specific definition of covariate shift does not affect our conclusions, nor does the type of shift under consideration. **We focus on covariate shifts because they are underexplored in the literature, and we believe selective classification provides a complementary approach towards robustness, which are often evaluated under covariate shifts.** Nonetheless, our results show that **our methods perform equally well on in-distribution data (ImageNet test set) and on semantic shifts (Table 7 in the appendix)**, underscoring the generality of the framework.
>
> > *Second, the practical implications of Lemma 2 are not clear.*
>
> **Response**: In Lemma 2, we assume that the true density of correctly (or wrongly) classified samples can be expressed as a multiplicative combination of the two component densities, $p_1$​ and $p_2$​, weighted by $\lambda$. The practical implication of such a distributional form is that **a sample is more likely to be correctly (or wrongly) classified when it has high likelihood under **both** $p_1$​ *and* $p_2$**. Under this view, a linear combination of the scores $s_1$ and $s_2$ **produces a more robust selector by leveraging complementary information from both distributions.** We will make this intuition clearer in the main text.
>
> As with our other theoretical results, **Lemma 2 is intended as a motivation for the linear combination strategy, and we do not claim that the underlying distribution truly follows this product form.** We will reiterate this in the revised manuscript. Other ways of combining scores are certainly possible, and we leave their exploration to future work.
>
> > *Third, Theorem 1 uses symbol "<<", which informally means "much smaller", but does not have any formal meaning*
>
> **Response**: The reviewer is correct that ‘$\ll$’ is exactly used to mean “much smaller” in this context. In Theorem 1,we use ‘$\ll$’ to express the assumption, which is utilized in the proof in Appendix A, that the majority of the softmax probability mass lies in the top two logits, i.e., $d^{(1)}$ and $d^{(2)}$. This behavior is commonly observed in modern classifiers, where the remaining logits collectively contribute only a small fraction of the probability mass.
>
> > *The newly introduced scores are not fundamentally new, since MDS and KNN scores have already been considered*
>
> **Response**: We agree with the reviewer that MDS and KNN themselves are not new, and we do not claim them as novel contributions. Rather, our contributions lie in: **(1) introducing a Neyman–Pearson–based framework that formalizes optimality in selective classification, (2) showing that this framework unifies existing post-hoc selector functions, and (3) proposing principled modifications to the MDS/KNN scores that lead to strong empirical improvements.**

---

> ### Author Response · Authors · 2025-11-19
> **Author response part 2**
>
> >  *On the practical side, it is not clear what amount of labelled data (e.g., relative to the amount of training data) is needed for the method to work reliably. This can be an issue, because modern classifiers can be constructed from pre-trained models with a minimum amount of training data (e.g., using few shot learning).*
>
> and
>
> > *What amount of labelled data (e.g., relative to the amount of training data) is needed for the method to work reliably?*
>
> **Response**: In the reported experiments, we follow established practices in OOD detection [1,2] and utilize the full ImageNet training set for estimating per-class means and covariances for $\Delta$-MDS.
>
> To investigate how our methods will behave in a few-shot setting, **we performed supplementary analyses on DFN CLIP in which the training set size was systematically reduced. The results show that both methods are surprisingly robust, with performance showing little change when using only 10% of the training samples.** Relative to the DFN-5B pretraining data (5B image-text pairs), this corresponds to only about 0.002% of the pretraining volume.
>
> Moreover, $\Delta$-KNN remains effective even under extremely constrained conditions. With only 0.1% of ImageNet (roughly 1K samples), $\Delta$-KNN exhibits minimal degradation, demonstrating that our approach can operate reliably with very limited compute and feature budgets. As expected, $\Delta$-MDS degrades at 1% due to the difficulty of estimating class-conditional statistics with so few samples, and is not applicable at 0.1% (roughly 1 sample per class). This distinction highlights $\Delta$-KNN’s advantage in low-sample regimes, as it does not rely on per-class statistics.
>
> Overall, these results indicate that our proposed methods, particularly $\Delta$-KNN, remain viable in the few-shot regime while being efficient in terms of computational requirements. We will include this discussion in the revised version.
>
> AURC/NAURC (lower is better) on ImageNet1K validation set.
> | Fraction       | 0.1%       | 1%           | 10%         | 50%         | 100%        |
> |----------------|-------------|--------------|-------------|-------------|-------------|
> | $\Delta$-MDS-RLog | -           | 10.5/0.638   | 4.19/0.202  | 4.14/0.198  | 4.13/0.197  |
> |  $\Delta$-KNN-RLog | 4.81/0.245  | 4.58/0.229   | 4.17/0.200  | 3.98/0.188  | 3.98/0.187  |
>
> > *In terms of presentation, it would be useful to define AURC within the paper.*
>
> **Response**: We thank the reviewer for this suggestion. We have defined the selective risk as $R(\phi) = \frac{E_{x,y}(\ell(f(x),y) \cdot g(x))}{\phi}$ in Eq. 4, where $\ell$ is the 0/1 loss, $g$ is the selector function and $\phi$ is the coverage. The AURC is simply the integral of $R$: $AURC = \int_0^1 R(\phi)d\phi$. We will include this definition in the revised manuscript.
>
> > *Also, I'm not sure how NP Lemma implies that "thresholding this score yields the lowest possible selective risk for any given coverage level"*
>
> **Response**: In our setting, the two hypotheses are “classifier classifies correctly” and “classifier classifies wrongly” with associated probabilities $p_c$ and $p_w$. A selector is therefore a hypothesis test that either accepts inputs or rejects (abstains). The Neyman-Pearson lemma states that thresholding the likelihood ratio $p_c/p_w$ minimizes the probability of accepting wrong points (type II error) *for any fixed constraint on rejecting correct points (type I error)*.
>
> Because the likelihood-ratio threshold determines the acceptance region, **fixing the type I error level is equivalent to fixing the coverage**. Since selective risk is the fraction of wrongly classified points among accepted points, this implies that thresholding (a monotone transform of) $p_c/p_w$ yields the lowest possible selective risk for any given coverage level. We hope this provides a clearer explanation, and we will incorporate it into the revised manuscript.
>
> > *The experiments do not provide confidence intervals or p-values*
>
> **Response**: We have not reported confidence intervals or p-values as the methods studied (both the baselines and our proposed scores) are fully deterministic, so repeated runs produce identical outputs. Moreover, the performance trends are consistent across many datasets, making it unlikely that the results arise by chance. Our reporting follows established practice in the selective classification and OOD detection literature [2,3].

---

> ### Author Response · Authors · 2025-11-19
> **Author response part 3**
>
> > *What do assumptions of Theorem 2 mean in practice?*
>
> **Response**: Theorem 2 assumes that the feature representations of correctly and wrongly classified samples are Gaussian, with separate per-class means and shared covariances. Under this condition, $\Delta$-MDS is Neyman–Pearson optimal.
>
> In practice, this assumption is motivated by prior work [1] showing that deep supervised classifiers trained with cross-entropy tend to yield features that are well-approximated by Gaussian Discriminant Analysis. Empirically, we see this reflected in our results: $\Delta$-MDS outperforms $\Delta$-KNN for EVA (a supervised model), whereas the opposite holds for CLIP, whose contrastive training does not align as well with the Gaussian assumption. As discussed in Sec. 5.1, a practical rule of thumb is to utilize $\Delta$-MDS and its variants with supervised classifiers, and $\Delta$-KNN for other settings.
>
> [1] Lee, Kimin, et al. "A simple unified framework for detecting out-of-distribution samples and adversarial attacks." Advances in neural information processing systems 31 (2018).
>
> [2] Yang, Jingkang, et al. "Openood: Benchmarking generalized out-of-distribution detection." Advances in Neural Information Processing Systems 35 (2022): 32598-32611.
>
> [3] Liang, Hengyue, Le Peng, and Ju Sun. "Selective Classification Under Distribution Shifts." arXiv preprint arXiv:2405.05160 (2024).
>
> Thank you for your feedback and we hope that we have addressed your concerns. If there are any further issues, please let us know.

---

> ### Comment · Reviewer_55hp · 2025-11-27
> **thanks**
>
> Thank you for the response which has addressed most of the concern. I will keep the score as is.

---

> > ### Author Response · Authors · 2025-11-28
> > **Thank you**
> >
> > We would like to thank the reviewer for their positive feedback!

---

### Official Review · Reviewer_z5TG · 2025-11-03

**Soundness:** 3
**Presentation:** 4
**Contribution:** 3
**Rating:** 8
**Confidence:** 3

**Summary:**

This paper tackles the problem of selective classification, where one has to decide whether to predict or abstain. The authors leverage the Neyman-Pearson lemma which sets up the problem as a hypothesis test between H0 (the classifier makes a correct prediction) and H1 (the classifier makes an incorrect prediction). The authors can cast existing methods such as RLog, MDS, and KNN into the Neyman-Pearson framework to show optimality. In the end, the method takes a linear combination of classifier logit scores and distance, which produce strong results on image and text classification benchmarks.

**Strengths:**

This paper provides a unified framework based on the Neyman-Pearson lemma that captures existing methods (which are often treated as ad-hoc).
The paper is fairly well-written and uses proper mathematical notation.
The empirical results are strong.

**Weaknesses:**

I think the optimality of Neyman-Pearson is a bit overstated, since optimality depends crucially on the distributional assumptions being valid.

**Questions:**

Methods like MDS require estimating the covariance, which could be statistically expensive (require many samples) compared to RLog? The methodology in the experiments could be a bit more transparent: how many examples are required? In the end, the hybrid methods improve over existing methods, but I want to make sure I understand the resources / additional tuning that's required for each one?

---

> ### Author Response · Authors · 2025-11-19
> **Author response**
>
> We thank the reviewer for acknowledging the strengths of our work and for their positive review. Below we provide our responses to the concerns and questions raised by the reviewer.
>
> > *I think the optimality of Neyman-Pearson is a bit overstated, since optimality depends crucially on the distributional assumptions being valid.*
>
> **Response**: We agree with the reviewer that, as with many theoretical assumptions in machine learning, the optimal conditions are idealized and are not expected to hold exactly in practice. We explicitly acknowledged this in Line 256: *“However, it does depend on asymptotic properties … may be difficult to satisfy in finite-sample settings. As such, both methods have their tradeoffs in terms of modeling assumptions”.*
>
> **Our intention is not to claim that these assumptions strictly hold in real-world scenarios, but to use them to motivate the design of our proposed scores and to clarify the principles underlying their construction**. Ultimately, the strongest evidence for the usefulness of our selectors comes from the empirical results, which demonstrate consistently strong performance across a wide range of experiments.
>
> On the Gaussian assumption of $\Delta$-MDS, we rely on prior work [1] that connects softmax classifiers with Gaussian Discriminant Analysis to justify this as a reasonable approximation. Nonetheless, we fully agree that feature distributions are not perfectly Gaussian in practice, and in the revised manuscript we will emphasize that this is an approximation rather than an exact description of the empirical distributions. Similarly, we have already stated that the asymptotic conditions on $\Delta$-KNN are not strictly achievable in practice.
>
> To avoid confusion, we will highlight more broadly that **these assumptions serve to illuminate the structure of an NP-optimal selector, rather than prescribe conditions that must hold in practice.** We welcome any further suggestions to ensure the framing remains balanced and appropriately calibrated.
>
> > *Methods like MDS require estimating the covariance, which could be statistically expensive (require many samples) compared to RLog? The methodology in the experiments could be a bit more transparent: how many examples are required? In the end, the hybrid methods improve over existing methods, but I want to make sure I understand the resources / additional tuning that's required for each one?*
>
> **Response**: We thank the reviewer for the thoughtful question. In our experiments, we follow established practices in OOD detection [1,2] and utilize the full ImageNet1K training set for estimating per-class means and the shared covariance for $\Delta$-MDS. This computation is performed once and can be cached, meaning its cost is fully amortized across all subsequent evaluations. RLog is a logit-based method and does not require feature computation, so no additional compute is introduced.
>
> **To address concerns about computational overhead and dependence on large training sets, we performed additional analyses on DFN CLIP to evaluate the robustness of our methods under reduced training set sizes. The results show that both methods are surprisingly robust, with performance changing little even when using only 10% of the training samples.**
>
> Importantly, $\Delta$-KNN remains effective even under extremely constrained settings. With just 0.1% of ImageNet (roughly 1K samples), $\Delta$-KNN exhibits minimal degradation, indicating that it can operate reliably with very limited compute and feature budgets. As expected, $\Delta$-MDS degrades at the 1% level due to the difficulty of estimating per-class statistics with so few samples, and is not applicable at 0.1% (roughly 1 image per class). This distinction highlights  $\Delta$-KNN’s advantage in low-sample regimes, as it does not rely on per-class statistics.
>
> Overall, these findings suggest that our proposed methods, particularly $\Delta$-KNN, remain practical, computationally efficient, and introduce minimal overhead. We will include this discussion in the revised manuscript.
>
> AURC/NAURC (lower is better) on ImageNet1K validation set.
> | Fraction       | 0.1%       | 1%           | 10%         | 50%         | 100%        |
> |----------------|-------------|--------------|-------------|-------------|-------------|
> | $\Delta$-MDS-RLog | -           | 10.5/0.638   | 4.19/0.202  | 4.14/0.198  | 4.13/0.197  |
> |  $\Delta$-KNN-RLog | 4.81/0.245  | 4.58/0.229   | 4.17/0.200  | 3.98/0.188  | 3.98/0.187  |
>
> [1] Lee, Kimin, et al. "A simple unified framework for detecting out-of-distribution samples and adversarial attacks." Advances in neural information processing systems 31 (2018).
>
> [2] Yang, Jingkang, et al. "Openood: Benchmarking generalized out-of-distribution detection." Advances in Neural Information Processing Systems 35 (2022): 32598-32611.
>
> Thank you for your positive review and we hope that we have addressed your remaining concerns. If there are any further issues, please let us know.

---

### Official Review · Reviewer_Vjb9 · 2025-11-03

**Soundness:** 3
**Presentation:** 2
**Contribution:** 2
**Rating:** 4
**Confidence:** 3

**Summary:**

This work studies the problem of selective classification where a classifier is allowed to abstain from making predictions if the model is not confident enough. This work proposed new approaches to selective classification  based on the Neyman-Pearson lemma and also unifies several existing approaches to this problem. They also provide experiments to support their theoretical results and show that their method outperforms various baselines under covariate shifts where the test input distribution is different form the train input distribution.

**Strengths:**

- The authors using NP lemma to combine several existing baseline methods is simple and intuitive.
- The authors proposed method - linear combination of distance based and logic based methods is simple and interesting.

**Weaknesses:**

- Theorem 2 relies on strong assumptions that the covariance distribution conditioned on the prediction is a gaussian. Theorem 3 relies on  k tending to infinity which is not practical.
- The authors do not provide intuitive understanding of in which cases, their proposed method should perform well compared to the baseline.

**Questions:**

- The authors in lemma 2 assume that density for each distribution takes a tilted form. Could the authors please elaborate that?
- The authors mention in line 305 that knn-distance based classifiers are ineffective on high dimensions. Why do they work for this work?
- Are the values of lambda and k similar across datasets and classifiers? Or, they have to be tuned separately for each setting?
- How is g^* computed in eqn 7?
- The authors have not defined SIRC method which is used as comparison in the results.

---

> ### Author Response · Authors · 2025-11-19
> **Author response part 1**
>
> We thank the reviewer for acknowledging the strengths our work and for their constructive comments. Below we provide our responses to the concerns and questions raised by the reviewer.
>
> > *Theorem 2 relies on strong assumptions that the covariance distribution conditioned on the prediction is a gaussian. Theorem 3 relies on k tending to infinity which is not practical.*
>
> **Response**: **Our intention is not to claim that these assumptions strictly hold in real-world scenarios, but to use them to motivate the design of our proposed scores and to clarify the principles underlying their construction.** Similar to many assumptions in ML theory, the optimality conditions may not hold exactly in practice and we explicitly acknowledged this in Line 256: *“However, it does depend on asymptotic properties … may be difficult to satisfy in finite-sample settings. As such, both methods have their tradeoffs in terms of modeling assumptions”.* Ultimately, the strongest evidence for the usefulness of our selectors comes from the empirical results, which demonstrate consistently strong performance across a wide range of experiments.
>
> Regarding the Gaussian assumption of $\Delta$-MDS, we rely on prior work [1] that connects softmax classifiers with Gaussian Discriminant Analysis to justify this as a reasonable approximation. Nonetheless, we agree that feature distributions are not perfectly Gaussian in practice, and in the revised manuscript we will emphasize that this is an approximation rather than an exact description of the empirical distributions. Similarly, we have already stated that the asymptotic conditions on $\Delta$-KNN are not strictly achievable in practice.
>
> To avoid confusion, we will highlight more broadly that **these assumptions serve to illuminate the structure of an NP-optimal selector, rather than prescribe conditions that must hold in practice.** We welcome any further suggestions to ensure the framing remains balanced and appropriately calibrated.
>
>
> > *The authors do not provide intuitive understanding of in which cases, their proposed method should perform well compared to the baseline.*
>
> **Response**: We thank the reviewer for the suggestion and will make this intuition more explicit in the revised version. As discussed in Sec. 5.1, a linear combination with $\Delta$-MDS tends to perform best for supervised classifiers, while a combination with $\Delta$-KNN is more suitable for self-supervised models like CLIP, whose feature distributions do not align well with Gaussian assumptions.
>
> Another practical consideration is data availability: our distance-based methods require a representative set of labeled samples. When such samples are available, combining our distance-based and logit-based scores works better than the baselines.
>
> **To illustrate this, we performed new experiments examining robustness of our methods to the number of labeled samples. The results (below) show that both methods are surprisingly stable with reduced sample sizes.** $\Delta$-KNN is especially robust, maintaining strong performance with as little as 0.1% of training data. As expected, $\Delta$-MDS degrades at the 1% level due to the difficulty of estimating per-class statistics with so few samples, and is not applicable at 0.1% (roughly 1 image per class). Importantly, $\Delta$-KNN-RLog continues to outperform RLog at 1% and matches it at 0.1% (see main paper), indicating that our method is still preferable whenever even a small amount of labeled data is accessible.
>
> AURC/NAURC (lower is better) on ImageNet1K validation set.
> | Fraction       | 0.1%       | 1%           | 10%         | 50%         | 100%        |
> |----------------|-------------|--------------|-------------|-------------|-------------|
> | $\Delta$-MDS-RLog | -           | 10.5/0.638   | 4.19/0.202  | 4.14/0.198  | 4.13/0.197  |
> |  $\Delta$-KNN-RLog | 4.81/0.245  | 4.58/0.229   | 4.17/0.200  | 3.98/0.188  | 3.98/0.187  |

---

> ### Author Response · Authors · 2025-11-19
> **Author response part 2**
>
> > *The authors in lemma 2 assume that density for each distribution takes a tilted form. Could the authors please elaborate that?*
>
> **Response**: The term “tilted” is borrowed from exponential tilting, and in our context, means a particular product form of distributions. In Lemma 2, we assume that the true density of correctly (or wrongly) classified samples can be expressed as a multiplicative combination of the two component densities, $p_1$​ and $p_2$​, weighted by $\lambda$. Intuitively, this means that **a sample is more likely to be correctly (or wrongly) classified when it has high likelihood under **both** $p_1$​ *and* $p_2$​**. Under this view, a linear combination of the scores $s_1$ and $s_2$ **produces a more robust selector by leveraging complementary information from both distributions.** We will make this intuition clearer in the main text.
> We emphasize that Lemma 2 is intended as a motivation for the linear combination strategy, and **we do not claim that the multiplicative product form is the true underlying distribution**. Other ways of combining scores are certainly possible, and we leave their exploration to future work
>
> > *The authors mention in line 305 that knn-distance based classifiers are ineffective on high dimensions. Why do they work for this work?*
>
> **Response**: We thank the reviewer for raising this point. Line 305 refers specifically to the Trust Score method, which relies on estimating class-conditional density level sets, a task that becomes unreliable in high-dimensional spaces. While a full comparison is beyond the scope of our work, we offer several hypotheses for why our KNN-based selector (and related KNN-based OOD methods [2]) performs well despite high dimensionality.
>
> First, modern neural networks provide substantially stronger and more structured representations than those used in Trust Score, making distances more meaningful. Second, our features are L2-normalized, so all points lie on the unit hypersphere, which simplifies distance comparisons in high dimensions. Third, we use the average log-distance, a smoother and better performing quantity than a single raw distance (as shown in our ablations). Most importantly, Trust Score requires estimating class-conditional density level sets, whereas our approach measures distances to the unconditional sets of correctly and incorrectly classified samples, arguably a simpler and more stable problem in high dimensions.
>
> > *Are the values of lambda and k similar across datasets and classifiers? Or, they have to be tuned separately for each setting?*
>
> **Response**: The values of $\lambda$ and $k$ used in our experiments are reported in appendix Table 5. While tuning these parameters for specific settings should yield the best results, our hyperparameter sensitivity analysis in appendix Figure 2 shows that our methods are relatively robust across a broad range of values.
>
> For $k$, we follow the ranges commonly used in the OOD detection literature [2], while for $\lambda$, our primary goal was to simply balance the magnitudes of $s_1$ and $s_2$ so that neither dominates the linear combination using a small number of training samples.
>
> > *How is g^* computed in eqn 7?*
>
> **Response**: In Eq. (7), the AURC with $g^\*$ is computed under an oracle (best possible) selector. Concretely, we assume the existence of a score function $s^\*$ such that **all** misclassified samples receive lower scores than **all** correctly classified samples. When thresholding $s^\*$, $g^\*$ therefore always rejects every wrongly classified sample *before* rejecting any correctly classified one, yielding the best possible risk–coverage tradeoff for that fixed classifier. **This oracle curve can be computed using the ground-truth labels on the evaluation set.** We will include this in the revision.
>
> > *The authors have not defined SIRC method which is used as comparison in the results.*
>
> **Response**: We thank the reviewer for catching this. We made a typo and referred to SIRC as SCOD on Line 366 of Sec. 5; we will correct this and provide a clearer description of all baselines in the revised version. In brief, SIRC is a method that combines multiple scores to jointly address OOD detection and selective classification. As noted in Sec. 4, it is primarily designed for semantic shifts and requires adaptation for covariate shifts. In contrast, our framework avoids this by representing all distribution shifts through the general pair $(p_c, p_w)$, which does not require distinguishing between shift types.
>
> [1] Lee, Kimin, et al. "A simple unified framework for detecting out-of-distribution samples and adversarial attacks." Advances in neural information processing systems 31 (2018).
>
> [2] Sun, Yiyou, et al. "Out-of-distribution detection with deep nearest neighbors." International conference on machine learning. PMLR, 2022.
>
> Thank you for your feedback and we hope that we have addressed your concerns. If there are any further issues, please let us know.

---

### Author Response · Authors · 2025-11-30
**Summary of Key Contributions and Responses**

We sincerely thank the reviewers for their constructive feedback and for the consensus that our Neyman-Pearson (NP) framework is a valuable contribution with strong empirical results.

To the new AC, thank you for taking on our paper. We summarize our key contributions and responses to the common feedback/queries below. We hope this will be helpful:

## Key Contributions:

1) A Neyman–Pearson framework that **formalizes optimality in selective classification** and provides principled guidance for selector design.
2) A **theoretical unification of existing post-hoc selector functions** as approximations to the NP-optimal likelihood ratio.
3) **NP-motivated modifications of MDS/KNN** that yield consistent empirical improvements across diverse distribution shifts.

## Main rebuttal points:

### 1) What is the sample complexity of the feature computation steps of our proposed methods (reviewers z5TG, 55hp)?

We added experiments varying the size of the reference feature set. We find that our methods are **effective down to 10% of training data**, with **$\Delta$-KNN matching baselines with as little as 0.1% of ImageNet**. These results demonstrate that our methods remain practical even in few-shot or resource-constrained settings. We also clarified that **feature statistics are computed once, making the costs minimal as it is amortized over all future evaluations**.

### 2) Strength of theoretical assumptions and optimality claims (all reviewers).

We clarified that the conditions, such as Gaussian and asymptotic KNN assumptions, are idealized and the **theory is used to motivate the score designs, not as requirements for practical deployment**. This aligns with most of machine learning theory, where strict adherence to stated assumptions are not required for the algorithms to be useful. **The strong empirical performance of our methods across diverse distribution shifts demonstrates the utility of our design principle.**

### 3) Practical implications of Lemma 2 (reviewers Vjb9, 55hp).

We explained that the “tilted” product-form assumption in Lemma 2 is a **modeling device to show why linear combinations of NP-optimal scores remain NP-optimal** and why they effectively fuse information from both scores, thus making it more robust. We do not assume the data necessarily follows this form; the usefulness of Lemma 2’s design is shown through **empirical results which show the linear combination scores achieving the best performance**.

### 4) Other clarifications (reviewers Vjb9, 55hp).
We corrected the covariate shift description and highlighted that **our formulation in terms of $(p_c, p_w)$ is agnostic to the type of distribution shift**, which is supported by experiments on both covariate and semantic shifts. We also **added practical guidance on when each selector is preferable** (e.g., $\Delta$-MDS for supervised models and $\Delta$-KNN for CLIP-style embeddings), and clarified evaluation details such as the **definition of AURC, how the oracle selector $g^\*$ is computed, hyperparameter selection and SIRC baseline**, among others.

Our individual responses below address the above points and others in detail. We believe that we have clarified all raised concerns, and we thank the reviewers again for their comments that have helped improve our work and to the AC for their efforts.

---

### Meta-Review · Area_Chair_Qm4a · 2026-01-03

**Summary:**

The reviewers all agree that the paper makes meaningful theoretical contribution to the selective classification domain. The common concerns/questions are about 1) framing of the optimality claim and practical implications in the context of required assumptions; 2) more clear analysis of sample complexity in experiments; 3) more discussions about comparison with related methods. Most of these are well accepted in the rebuttal. The authors are encouraged to carefully incorporate these the suggestions into their revision.

**Reviewer Concerns:**

In the rebuttal, authors calibrated their optimality claim. They also provided more analysis about the experiments, more experiments comparing with related methods, and discussion of practical implications of their theory.

**Reviewer Scores:**

Reviewer 55hp mentioned that they will keep the score 6.

I predict Reviewer z5TG will keep the positive score after the rebuttal.

 Reviewer Vjb9 may raise their score, given the rebuttal.

---

### Decision · Program_Chairs · 2026-01-26

Accept (Poster)